# Flatness is Necessary, Neural Collapse is Not: Rethinking Generalization via Grokking

**Ting Han**
Lamarr Institute, TU Dortmund, Germany
and Institute for AI in Medicine, UK Essen
`ting.han@tu-dortmund.de`

**Linara Adilova**
Research Center Trustworthy Data Science
and Security of the University Alliance Ruhr,
TU Dortmund, Germany

**Henning Petzka**
Ruhr University Bochum, Germany

**Jens Kleesiek**
Institute for AI in Medicine, UK Essen
and Department of Physics, TU Dortmund, Germany

**Michael Kamp**
Lamarr Institute, TU Dortmund, Germany
and Institute for AI in Medicine, UK Essen
`michael.kamp@tu-dortmund.de`

## Abstract

Neural collapse, i.e., the emergence of highly symmetric, class-wise clustered representations, is frequently observed in deep networks and is often assumed to reflect or enable generalization. In parallel, flatness of the loss landscape has been theoretically and empirically linked to generalization. Yet, the causal role of either phenomenon remains unclear: Are they prerequisites for generalization, or merely by-products of training dynamics? We disentangle these questions using grokking, a training regime in which memorization precedes generalization, allowing us to temporally separate generalization from training dynamics and we find that while both neural collapse and relative flatness emerge near the onset of generalization, only flatness consistently predicts it. Models encouraged to collapse or prevented from collapsing generalize equally well, whereas models regularized away from flat solutions exhibit delayed generalization, resembling grokking, even in architectures and datasets where it does not typically occur. Furthermore, we show theoretically that neural collapse leads to relative flatness under classical assumptions, explaining their empirical co-occurrence. Our results support the view that relative flatness is a potentially necessary and more fundamental property for generalization, and demonstrate how grokking can serve as a powerful probe for isolating its geometric underpinnings.[1]

## 1 Introduction

Overparameterized neural networks continue to challenge classical learning theory. Despite their capacity to memorize arbitrary labels [Zhang et al., 2017], they often generalize well on natural data and, in some cases, only begin to generalize after complete memorization, a phenomenon known as grokking [Power et al., 2022]. This apparent contradiction has motivated a renewed interest in identifying geometric signatures of generalization. Two candidates have emerged as particularly prominent: the flatness of the loss landscape [Keskar et al., 2017, Jiang et al., 2020, Petzka et al.,

---

[1]Code implementation: https://github.com/TrustworthyMachineLearning-Lab/grokking_flatness.

39th Conference on Neural Information Processing Systems (NeurIPS 2025).

2021] and the neural collapse (NC) phenomenon [Papyan et al., 2020, Mixon et al., 2022, Zhou et al., 2022]. Both tend to appear late in training and are frequently associated with good generalization, yet their precise causal roles remain unclear.

In this paper, we challenge the common conjecture that NC is necessary for generalization [Mixon et al., 2022, Zhou et al., 2022, Súkeník et al., 2023, Zhu et al., 2021]. Our key idea is to leverage grokking as a unique observational window: since generalization emerges only after prolonged memorization, it enables a clean separation between training dynamics and generalization. This allows us to ask: do NC and flatness merely correlate with generalization, or do they contribute to it?

We measure both phenomena in the penultimate layer. For flatness, we adopt the relative flatness metric proposed by Petzka et al. [2021], which considers the Hessian trace normalized by the weight norm, a quantity theoretically and empirically shown to align with generalization. To make this computable in large state-of-the-art neural networks, we employ the alternative closed-form upper bound introduced by [Walter et al., 2024], which is valid in the penultimate layer under cross-entropy loss. For NC, we measure the class-wise clustering of penultimate-layer representations through empirical variance and mean properties, tracking its emergence throughout training. In grokking experiments, we find that flatness emerges alongside generalization, never before. NC, in contrast, appears earlier during memorization, suggesting that it can lead to, but is not neccessary for generalization (cf. Fig. 1).

To isolate their functional roles, we perturb training dynamics. When we suppress collapse-style clustering, networks still generalize robustly; test accuracy remains high despite high clustering values. Flatness, however, is unaffected. In contrast, regularizing against flatness, i.e., encouraging sharp solutions, reliably delays generalization and induces grokking-like behavior even in settings where it does not typically arise, including ResNets on CIFAR-10, ViTs on ImageNet and various pre-trained language models on SST-5. These findings point to flatness as a more fundamental driver of generalization. To explain the frequent co-occurrence of NC and flatness, we provide a theoretical result: under classical NC assumptions, collapse *leads to* relative flatness. This unifies previously observed correlations under a single geometric framework, but also shows that flatness can arise via other mechanisms. NC is thus a sufficient but non-exclusive path toward flatness. Finally, we revisit the theoretical framework of Petzka et al. [2021], which as-

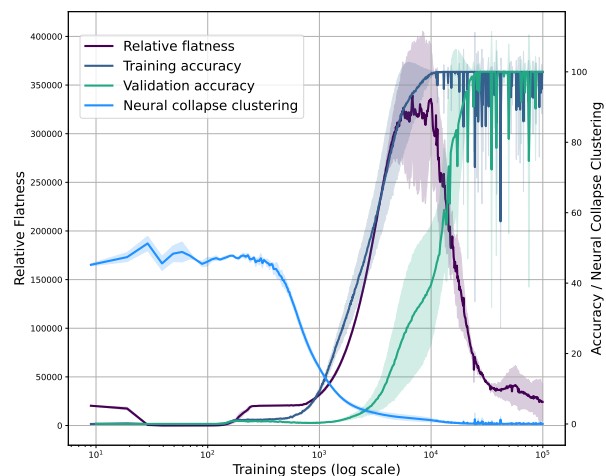

Figure 1: Neural collapse clustering and relative flatness in grokking. While both correlate with generalization, neural collapse emerges early during memorization, whereas flatness only drops sharply when generalization begins, highlighting flatness as a better indicator of generalization onset.

serts that generalization requires not only flatness, but also *representativeness*, i.e., the alignment between learned features and the true data distribution. Our experiments confirm this dependency. Representativeness, when approximated, improves only as generalization sets in, highlighting that neither flatness nor collapse alone ensures generalization without meaningful feature alignment.

Together, these results place NC in a new light. While it often accompanies generalization, it is not a prerequisite. Instead, it may serve as an inductive bias toward flatness under standard training dynamics. Relative flatness, by contrast, tends to be empirically necessary and theoretically justified, especially when labels are locally constant and features are representative—assumptions which appear to hold in our experimental settings.

Our work provides new insight into the geometry of generalization:

- We show empirically that relative flatness, but not neural collapse, is potentially necessary for generalization;
- we prove that neural collapse leads to relative flatness under classical assumptions;
- we demonstrate that grokking-like behavior can be actively induced by manipulating flatness, even in real-world architectures.

These insights challenge prevailing narratives about generalization in deep learning and suggest new levers for training, regularization, and model diagnostics.

## 2 Related Work

Despite substantial empirical success, the mechanisms driving generalization in deep neural networks remain poorly understood. Two pronounced geometric perspectives, *neural collapse* and *loss surface flatness*, have emerged as phenomena empirically correlated with generalization, but their interrelation and causal roles remain ambiguous. In this work, we leverage the properties of grokking to disentangle necessity from sufficiency and to assess which of these phenomena are required for generalization.

**Neural Collapse and Generalization**    *Neural collapse* (NC) describes a geometric convergence during training a classifier where penultimate layer features from the same class collapse to a single mean, the means form a simplex equiangular tight frame, and classifier weights align accordingly [Papyan et al., 2020]. This phenomenon has been widely observed at late training stages in overparameterized networks [Han et al., 2022, Graf et al., 2021, Rangamani and Banburski-Fahey, 2022, Wu and Papyan, 2024, Zhou et al., 2025] and has become a widely accepted indicator for generalization and robustness. Under idealistic assumption of an unconstrained features model, i.e., a model which optimizes features as free variables, NC has been shown to be the global optimal solution for various loss functions [Mixon et al., 2022, Zhou et al., 2022, Súkeník et al., 2023, Zhu et al., 2021]. Similar results for compatibility of zero test error and collapse of variances (so only NC1 condition) were shown for mean-field networks [Wu and Mondelli, 2025]. Practical applications have leveraged NC properties for downstream performance. For instance, Wu et al. [2025] use a feature separation loss based on NC to improve out-of-distribution detection. Similarly, Munn et al. [2024] show that lower geometric complexity in pre-trained representations fosters NC and leads to better few-shot generalization in transfer learning. Galanti et al. [2022] shows theoretically and empirically that NC is the reason for transfer learning to work better. These findings suggest that NC may be *sufficient* for generalization in certain regimes. However, the open question remains: is NC *necessary* outside of the unconstrained features models? For example, NC is a debated matter in the context of inbalanced learning, where it was observed that minority collapse can happen, i.e., the means of the minority classes merge, thus preventing classification [Fang et al., 2021]. At the same moment it is possible to induce NC by enforcing it through training, even in the unbalanced data setup [Yang et al., 2022]. For deep linear networks it was also theoretically shown that the global optimum exhibits NC, but even more: Under unbalanced data, NC gets means proportional to the class size [Dang et al., 2023]. Hui et al. [2022] argues for measuring NC on the test set for understanding generalization properties and demonstrates that test set NC is hurting generalization performance on related downstream tasks, i.e., representations with strong NC on the test set are bad for transfer learning. Moreover, Hong and Ling [2024] show that a sufficiently high signal-to-noise ratio is a necessary condition for NC to correlate with good generalization, explaining why strong NC can otherwise coincide with poor transfer performance.

Overall, the existing research suggests that NC is common, but might not be required for generalization: a network can generalize without NC and some setups (like noisy labels) can break the link between generalization and NC. Our work addresses this question directly by constructing settings in which models generalize without exhibiting NC, demonstrating that it is not a prerequisite for generalization.

**Flatness and Generalization**    The connection between loss surface flatness and generalization has long been suspected [Hochreiter and Schmidhuber, 1997], and more recently has been supported by empirical studies showing that flatness-based measures often outperform alternatives such as norm-, margin- or optimization-based metrics [Jiang et al., 2020, Keskar et al., 2017]. However, classical flatness measures, typically based on the Hessian of the loss with respect to parameters,

are known to be sensitive to reparameterizations that leave the function and generalization behavior unchanged [Dinh et al., 2017], and computationally expensive for large state-of-the-art models. To resolve this, Petzka et al. [2021] introduced a reparameterization-invariant measure of *relative flatness*, grounded in a theoretical framework that connects flatness of an individual layer to generalization under assumptions of representative data and locally constant labels. This formulation not only explains prior empirical correlations, but also provides theoretically grounded conditions under which flatness predicts generalization. Walter et al. [2024] further made this measure more practical by deriving closed-form computation formulas in standard classification settings. In our work, we empirically verify that relative flatness drops precisely when generalization emerges during grokking. Moreover, we demonstrate that promoting relative flatness can actively *induce* generalization even in the absence of NC. In doing so, we confirm and extend the theoretical framework of Petzka et al. [2021], while disentangling flatness from NC.

**Grokking as a Window into Generalization**   *Grokking* refers to the sudden emergence of generalization long after perfect training performance is achieved, as first described by Power et al. [2022]. This phase transition, which is often induced by implicit or explicit regularization, offers a unique opportunity to isolate potential causes for the emergence of generalization. Recent work explains grokking through two-phase dynamics: an initial kernel regime focused on memorization, followed by feature learning that supports generalization [Mohamadi et al., 2023, Lyu et al., 2024, Kumar et al., 2024]. Analytical studies reveal closed-form grokking solutions in modular arithmetic tasks [Gromov, 2023, Žunkovič and Ilievski, 2024], while mechanistic studies identify sparse and dense subnetworks competing during training [Nanda et al., 2023, Gouki et al., 2023]. Most of these studies focus on shallow networks or toy tasks. However, recent work shows that grokking also arises in deeper architectures and real-world datasets [Murty et al., 2023, Humayun et al., 2024], suggesting greater relevance. We build on this foundation by using grokking as an experimental lens: By tracking the timing of flatness, NC, and generalization, we are able to distinguish which factors are causally connected and which are merely correlated.

## 3   Geometric Foundations of Generalization

In this work, we investigate the interplay between generalization and two notable phenomena in modern deep learning: *neural collapse*, and *flatness* of the loss surface. These concepts are frequently observed to co-occur late in training, but their individual roles and causal relations remain poorly understood. Our goal is to disentangle their contributions to model performance by leveraging the grokking setting, in which memorization precedes generalization, allowing us to observe the emergence of each phenomenon in isolation.

**Generalization**   A central objective in machine learning is to ensure that a trained model not only performs well on the training data, but also generalizes to unseen data drawn from the same distribution. This difference in performance is formalized via the *generalization gap*. Let $f : \mathcal{X} \to \mathcal{Y}$ be a model from a model class $\mathcal{F}$ trained with a twice-differentiable loss function $\ell : \mathcal{Y} \times \mathcal{Y} \to \mathbb{R}_+$ on a finite training set $S \subseteq \mathcal{X} \times \mathcal{Y}$, drawn i.i.d. from a data distribution $\mathcal{D}$ over the input space $\mathcal{X}$ and output space $\mathcal{Y}$. The generalization gap is defined as $\mathcal{E}_{\text{gen}}(f, S) := \mathcal{E}(f) - \mathcal{E}_{\text{emp}}(f, S)$, where $\mathcal{E}(f) := \mathbb{E}_{(x,y)\sim\mathcal{D}}\left[\ell(f(x), y)\right]$ is the risk and $\mathcal{E}_{\text{emp}}(f, S) := \frac{1}{|S|}\sum_{(x,y)\in S}\ell(f(x), y)$ is the empirical risk. The model generalizes well when $\mathcal{E}_{\text{gen}}(f, S)$ is small, indicating close alignment between training and test performance.

**Neural Collapse and the NCC Measure**   *Neural collapse* (NC) is a geometric phenomenon characterizing the late stages of training in deep classification networks [Papyan et al., 2020, Mixon et al., 2022]. It manifests as an alignment of the learned representations such that: (1) within each class, features in the penultimate layer collapse to their class mean; (2) the class means themselves form a simplex equiangular tight frame, a maximally spaced configuration; (3) the classifier weights align with these class means; and (4) the classifier effectively becomes a nearest-neighbor model.

This structure has been theoretically motivated and empirically observed in both training and unseen test samples, including new classes, particularly in transfer learning settings [Galanti et al., 2022]. To quantify the emergence of NC, we adopt the simplified Neural Collapse Clustering (NCC) measure proposed by Galanti et al. [2022], which captures the essential characteristics of neural collapse, i.e.,

tight intra-class clustering and strong inter-class separation, without requiring all four formal NC conditions. This is the measure we use throughout our analysis.

**Definition 3.1** (NCC Measure). Let $\phi(x)$ denote the penultimate-layer representation of input $x$, and $D_c$ the set of training samples from class $c$. Then

$$\text{NCC} := \sum_{c \neq c'} \frac{V_c + V_{c'}}{2\|\mu_c - \mu_{c'}\|^2},$$

where $\mu_c := \frac{1}{|D_c|} \sum_{x \in D_c} \phi(x), V_c = \sum_{x \in D_c} \|\phi(x) - \mu_c\|^2$ are the mean, respectively the variance of representations of class $c$.

A low NCC value indicates tight intra-class clustering and well-separated class means, characteristic of NC. In Section 5, we confirm that the NCC measure correlates with the angular separation of means, thus capturing all the characteristics of NC.

**Relative Flatness**  Flatness of the loss landscape, understood as the insensitivity of the training loss to small perturbations in the parameter space, has long been linked to generalization research [Hochreiter and Schmidhuber, 1997, Keskar et al., 2017]. However, classical measures based on the Hessian or curvature are sensitive to reparameterizations, making them unreliable in modern architectures. To overcome this, Petzka et al. [2021] introduced a reparameterization-invariant notion of *relative flatness*, grounded in a theory of robust generalization under locally constant labels.

Consider a model that decomposes as $f(x, \mathbf{w}) = g(\mathbf{w}\phi(x))$, where $\phi$ is a fixed feature map (e.g., the penultimate layer), $\mathbf{w} \in \mathbb{R}^{d \times m}$ is a weight matrix, and $g$ is a twice-differentiable function. The relative flatness is defined via the trace-weighted Hessian along the directions spanned by $\mathbf{w}$.

**Definition 3.2** (Relative Flatness). Let $g, \ell$ be twice differentiable, and $S$ a sample set. Then:

$$\kappa_{\text{Tr}}^{\phi}(\mathbf{w}) := \sum_{s,s'=1}^{d} \langle \mathbf{w}_s, \mathbf{w}_{s'} \rangle \cdot \text{Tr}(H_{s,s'}(\mathbf{w}, \phi(S))),$$

where $\mathbf{w}_s$ denotes the $s$-th row of $\mathbf{w}$, $\langle \cdot, \cdot \rangle$ is the scalar product, and $H_{s,s'}(\mathbf{w}, \phi(S))$ is the Hessian of the empirical loss with respect to $\mathbf{w}_s$ and $\mathbf{w}_{s'}$ evaluated at $\phi(S)$.

This measure is invariant to neuron-wise rescaling and composition with orthogonal transformations, making it robust across different parameterizations. Under mild assumptions—namely, that the training data is representative and the labels are locally constant in feature space—relative flatness predicts generalization. Note that a small value of $\kappa_{\text{Tr}}^{\phi}(\mathbf{w})$ is indicative of a flat solution and good generalization. In Section 5, we further show that NC leads to low relative flatness, establishing a theoretical dependency between the two phenomena.

**On the Role of Representativeness**  Both neural collapse and relative flatness have been proposed as geometric signatures of generalization. However, neither property is meaningful in isolation: their predictive value depends critically on the relationship between training and test distributions in feature space. In particular, if the training features are not representative of the full data distribution, then both NC and flatness can emerge without leading to generalization. For example, consider a network that perfectly collapses the training set into a class-wise simplex configuration and assigns test samples to incorrect cluster means. (On discrete data, such a network function can be easily constructed.) Prediction will then fail despite the presence of collapse. This limitation also holds for relative flatness: a flat but non-representative model does not generalize [cf. Sec. 2, Petzka et al., 2021]. While NC lacks a rigorous definition of representativeness, relative flatness is grounded in a formal framework that makes the required assumptions explicit. In particular, the theory of *ε-representativeness* [Petzka et al., 2021] defines a condition under which robustness in feature space becomes necessary for generalization. Under this assumption, relative flatness serves as an upper bound on feature robustness and provides a tractable diagnostic for generalization. This perspective clarifies that our analysis operates under the assumption of representative features—an assumption that is empirically supported in our settings, as we demonstrate in Appendix D.

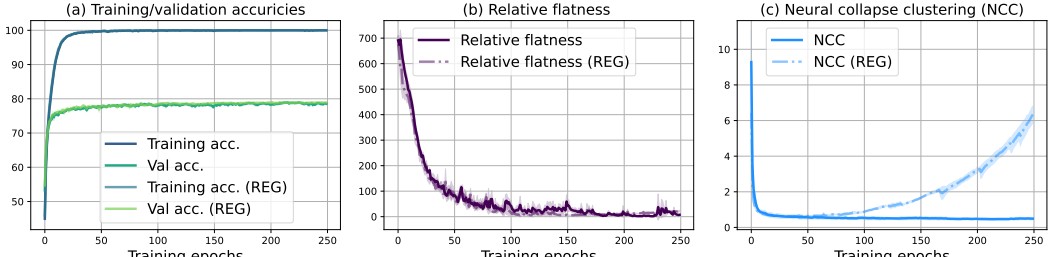

Figure 2: Results of neural collapse clustering regularization on CIFAR-10. We display both unregularized and regularized (REG) training dynamics for comparison. Increasing NCC does not affect generalization or relative flatness, indicating that NC is not necessary for generalization. Figure (a) shows the training and validation accuracies, and y-axis represents accuracy. Figure (b) presents the relative flatness values during training, and y-axis represents measurement of relative flatness. Figure (c) illustrates the NCC development, and y-axis represents NCC value.

## 4 Neural Collapse and Flatness in Delayed Generalization

Grokking refers to the surprising phenomenon in which a neural network, after memorizing the training set with zero generalization for an extended period, suddenly begins to generalize with high accuracy, often after tens or hundreds of thousands of additional optimization steps [Power et al., 2022]. This temporal separation between memorization and generalization provides a unique opportunity to disentangle the factors that underlie generalization. In our setting, we use grokking, or delayed generalization, not as a curiosity, but as a causal probe: a mechanism for identifying which geometric properties emerge coincidentally with generalization, and which ones may be functionally necessary for it.

First, we study grokking on symbolic algorithmic tasks such as modular arithmetic (e.g., $x + y \mod p$), where networks are trained to predict the output of binary operations over abstract tokens. These tasks are small, fully observable, and require exact generalization beyond memorized samples, making them an ideal testbed. We follow the original setup of Power et al. [2022], training a 2 layer-transformer using the AdamW optimizer with a learning rate of $10^{-4}$ and weight decay of 1.0. Each experiment runs for $10^6$ steps with a 50/50 train/validation split. All results are averaged over three seeds. Figure 1 shows the evolution of train and validation accuracy, as well as relative flatness (Def. 3.2), and neural collapse clustering (Def. 3.1) throughout training. We observe that both measures decrease sharply when the network starts generalizing, however, NCC already decreases during the memorization phase. Relative flatness remains high during the memorization phase and decreases sharply near the point at which generalization begins. This temporal coincidence suggests that both NCC and relative flatness are correlated with generalization, but representations start exhibiting collapse behaviour before generalization has emerged.

However, temporal correlation is not causation. The simultaneous decrease of NCC and flatness may reflect shared dependence on generalization, or confounders, rather than causal influence. The grokking setting thus makes it clear that we must go further: it is not enough to ask whether these quantities track generalization, we must ask whether they are causing it. In the following sections, we address this question empirically. In Section 5, we show that neural networks can generalize without exhibiting collapse. In Section 6, we provide empirical evidence that relative flatness is necessary for generalization under mild assumptions.

## 5 Neural Collapse is not Necessary

Neural collapse (NC) is often observed in models that generalize well, leading to the impression that it may play a functional role in generalization. However, correlation does not imply necessity. In this section, we demonstrate both empirically and theoretically that NC is not required for generalization. While collapse can accompany good generalization, it is not necessary: It is one path that leads to flatness, and through flatness, to generalization.

To substantiate this claim, we explicitly suppress the emergence of NC during training without affecting generalization. We train a ResNet-18 on CIFAR-10 using a regularized loss of the form:

$$\mathcal{L}_{\text{NC\_REG}} = \mathcal{L}_{\text{CE}} - \lambda \cdot \text{NCC},$$

where $\mathcal{L}_{\text{CE}}$ is the standard cross-entropy loss and $NCC$ is the NC clustering measure (Def. 3.1). The NCC term penalizes tight clustering of penultimate-layer features by shrinking the distances between class means and increasing intra-class variance, thereby actively discouraging the geometric structure characteristic of NC. The training hyperparameter setups, additional experimental details are provided in Appendix B.

We show that generalization can occur without NC in Figure 2. As shown in panel (a), both training and validation accuracies remain unaffected by the regularizer. Panel (b) shows that relative flatness, our proxy for generalization, also remains stable. However, panel (c) reveals that NC is effectively suppressed: after a brief drop in the NCC measure, it steadily increases throughout training. This provides clear evidence that generalization can occur in the complete absence of NC, and that collapse is not a necessary condition for generalization. In Figure 3 we confirm that the NCC measure correctly captures cluster angles as well. Under neural collapse, angles are close to the optimal 10-simplex, yet when we regularize with the NCC measure, angles deviate strongly from this optimum.

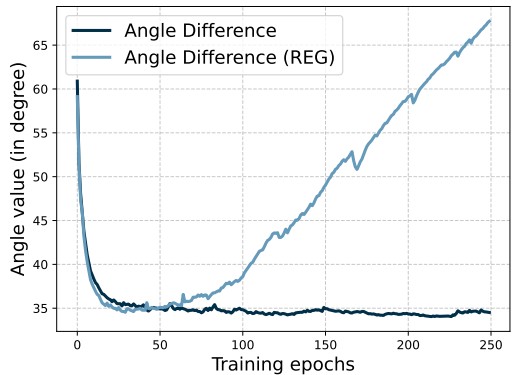

Figure 3: Pairwise cluster angles vs. optimal 10-simplex angles. Under NCC regularization, angles drift from the optimal configuration, while in standard training they remain stable. "REG" indicates use of the NCC regularizer.

Why, then, does neural collapse often accompany generalization? We argue that collapse is one geometric pathway to flat solutions, which are more directly linked to generalization. In fact, under mild assumptions, NC in the penultimate layer leads to relative flatness[2]. This relationship is formalized below.

We assume that classical neural collapse [Papyan et al., 2020] in the limit holds for a network function $f$. That is, the following four conditions apply to the neural network $f(x) = \text{softmax}(w\phi(x) + b)$ with $\phi(x)$ as the representation of the penultimate layer.

**Assumptions 5.1** (Neural Collapse [Papyan et al., 2020]). Let $\phi(x)$ denote the penultimate layer feature representation of an input $x \in \mathcal{X}$, and for a dataset $D \subset \mathcal{X}$, let $D_c \subseteq D$ denote the set of inputs belonging to class $c \in \{1, \ldots, k\}$. Then the four neural collapse criteria are

(NC1) Feature representations collapse to class means: $\phi(x) = \mu_c$ for all $x \in D_c$.

(NC2) All class means $\mu_c$ have equal distance to the global mean $\mu_g$, i.e., $\|\mu_c - \mu_g\|_2 = M$, and they form a centered equiangular tight frame (ETF), so that $(\mu_j - \mu_g)^\top (\mu_c - \mu_g) = -\frac{M^2}{k-1}$ for $j \neq c$.

(NC3) The classifier weights align with class means, i.e., $w_j = \lambda(\mu_j - \mu_g)$ for all $j$ and some $\lambda > 0$.

(NC4) $\text{argmax}_j w_j^\top h + b_j = \text{argmin}_j \|h - \mu_j\|$.

*Remark* 5.2. For the proof of Proposition 5.3, we additionally assume that $\|\mu_c\| \leq M$ for each $c$. This entails that the center of the ETF is not diverging. This mild technical condition does not alter the essence of the neural collapse assumptions.

---

[2]Related works often rely on simplifying assumptions, e.g., one-layer behavior or layer peeling [Mixon et al., 2022, Fang et al., 2021], yet our analysis does not require such approximations.

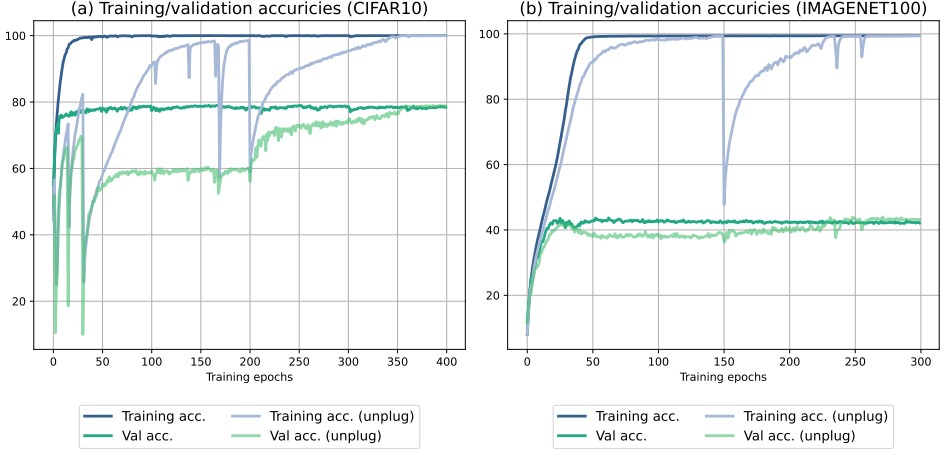

Figure 4: Results of inducing delayed generalization on training Resnet18 on CIFAR-10 and ViT on ImageNet-100 through relative flatness regularization. We display both unregularized and regularized (unplug) training dynamics for comparison. Relative flatness regularizer is removed at epoch 200 for CIFAR-10 experiment and 150 for ImageNet-100. Delayed generalization occurs after the regularizer is removed, as indicated by sharp increase in validation accuracy. Figure (a) shows the training and validation accuracies on CIFAR-10, and Figure (b) on ImageNet-100.

**Proposition 5.3.** *Let $f(x) = softmax(w\phi(x) + b)$ be a neural network with softmax output and trained with cross-entropy loss, where $w \in \mathbb{R}^{k \times d}$ denotes the final-layer weight matrix classifying into $k$ classes and $\phi(x)$ is the penultimate-layer representation. Assume that the classical neural collapse holds in the limit as above. Then relative flatness $\kappa_\phi(w)$ is bounded by:*

$$\kappa_\phi(w) \leq \lambda^2 k^3 M^4 \cdot \frac{e^{-\lambda M^2 \cdot \frac{k}{k-1}}}{\left(1 + (k-1)e^{-\lambda M^2 \cdot \frac{k}{k-1}}\right)^2}$$

*In particular, for sufficiently large $\lambda$, this yields the asymptotic bound:*

$$\kappa_\phi(w) \lesssim \lambda^2 k^3 M^4 e^{-\lambda M^2 \cdot \frac{k}{k-1}},$$

*which decays exponentially in $\lambda$.*

The proof is provided in Appendix A. The fact that relative flatness indeed decreases to zero in the limit follows from the fact that the conditions of neural collapse are independent of the weight norm and thereby also the magnitude of the scalar $\lambda$. As training continues in the NC limit, the training loss can be decreased to zero by increasing $\lambda$, because the softmax probabilities converge to $\hat{y}_c(x_c) = 1$ and $\hat{y}_j(x_c) = 0, j \neq c$ as $\lambda \to \infty$.

Taken together, our results demonstrate that, while NC can facilitate generalization by inducing flatness, it is not a prerequisite. Flatness remains the more fundamental quantity, and its relevance critically depends on the representativeness of the learned feature space. It remains to investigate whether flatness is a sufficient condition and whether it could be necessary.

## 6 Relative Flatness is Necessary

In this section, we induce delayed generalization, similar to what is observed in grokking. These experiments demonstrate that relative flatness is necessary for generalization. To achieve such behaviour, we design a loss function that includes the following regularization term:

$$\mathcal{L}_{\text{RF\_REG}} = \mathcal{L}_{\text{CE}} - \lambda \cdot \kappa_{Tr}^\phi(\mathbf{w}),$$

where $\kappa_{Tr}^\phi(\mathbf{w})$ is specified in Definition 3.2, and $\mathcal{L}_{\text{CE}}$ denotes the cross-entropy loss. To efficiently compute the relative flatness on the penultimate layer, we adopt the closed-form derivation under

cross-entropy loss provided by Walter et al. [2024]. By maximizing the relative flatness, we encourage the network to explore sharper regions of the loss landscape rather than flatter ones, which leads to suboptimal generalization. The coefficient $\lambda$ controls the strength of the regularization. We train ResNet-10 on CIFAR-10 and ViT [Dosovitskiy et al., 2021] on ImageNet-100[3], both from scratch, to evaluate the effectiveness of inducing delayed generalization in general settings by regularizing relative flatness. The results are shown in Figure 4. We also fine-tune TinyBERT [Jiao et al., 2020] and DistilGPT2 [Sanh et al., 2019] on SST-5 to show that delayed generalization can be induced by regularizing relative flatness in pretrained language models. It is worth noting that stabilizing training in sharp minima under the relative-flatness regularizer is nontrivial, since cross-entropy loss reduces prediction uncertainty (in terms of the entropy of softmax), whereas the regularizer operates effectively only under high uncertainty. Additional analysis of the behavior and sensitivity of the regularizer, particularly its conflict with cross-entropy loss, is provided in Appendix C, which also includes the training setups and additional experimental details.

From Figure 4 (a), we observe that the validation accuracy is low, but increases after the relative flatness regularizer is removed at epoch 200. A similar phenomenon is observed in the results on ImageNet-100, shown in Figure 4 (b) and in the NLP experiments presented in Appendix C.4. This recovery is facilitated by applying standard regularization techniques after unplugging, helping the model escape sharp minima and restore validation accuracy to the baseline. The successful induction of delayed generalization in these tasks demonstrates that this effect is not exclusive to algorithmic tasks trained with the next-token prediction objective. By surpressing relative flatness, we can induce delayed generalization in more general tasks and training settings.

In particular, when the relative flatness regularizer is applied throughout the entire training process, the final validation accuracy remains around $60\%$ on CIFAR-10, whereas the optimal validation accuracy reaches approximately $78\%$. For ImageNet-100, the stable validation accuracy under regularization is around $38\%$, compared to an optimal accuracy of around $43\%$. A detailed analysis of pretrained language models on SST-5 is provided in Appendix C.4. This gap in validation accuracy, caused by the absense of relative flatness, suggests that optimal generalization cannot be achieved in such a way.

To further support the need for solutions to be flat for better generalization, we compute the relative flatness values at the epoch right before the regularizer is removed and at the final training epoch. For CIFAR-10, at the epoch 199, the measure of relative flatness under regularization is 2209.94, which decreases to 1.91 at the final epoch. For ImageNet-100, the measure is 22212.75 at epoch 149 and 313.22 at the final epoch. We also observe similar trends on SST-5 with TinyBERT and DistilGPT2 (Appendix C.4). Therefore, a smaller generalization gap is directly correlated with the value of relative flatness.

Overall, these results provide strong empirical evidence that *relative flatness is necessary for generalization*, consistently observed across all evaluated tasks.

# 7 Discussion and Conclusion

Our results identify relative flatness as a potentially necessary condition for generalization in deep networks, while neural collapse emerges as a correlated but non-essential feature of late-stage training. This distinction reshapes our understanding of the geometry of well generalizing solutions and invites several important reflections. Our experiments show that neural collapse tends to coincide with high training accuracy but does not reliably track generalization. This suggests that it is more a byproduct of training dynamics than a causal mechanism, a view that aligns with findings from Hui et al. [2022] and Hong and Ling [2024]. Nevertheless, we show theoretically that under mild assumptions, neural collapse-style clustering leads to relative flatness. As such, collapse may still serve as a practical proxy for flat solutions in some settings.

While neural collapse produces highly symmetric and compact feature representations, this structural simplicity can be problematic in transfer learning and out-of-distribution scenarios. For instance, if a network is trained to classify supercategories by collapsing fine-grained class distinctions, it becomes impossible to recover those distinctions later, thus limiting the generality and interpretability of the learned features [Hui et al., 2022]. The requirement for geometric compression of the representation

---

[3]https://www.kaggle.com/datasets/ambityga/imagenet100/data

space, with neural collapse as one instance, has also been questioned from an information-theoretic perspective on deep learning dynamics [Adilova et al., 2023b].

Flatness itself has also come under scrutiny, particularly in the context of sharpness-aware minimization (SAM) [Foret et al., 2021] and recent critique by Andriushchenko et al. [2023]. However, many such criticisms target classical flatness metrics, which are sensitive to reparameterization [Dinh et al., 2017] or measures with unclear connection to geometric flatness [Andriushchenko and Flammarion, 2022]. Our analysis relies on relative flatness, a reparameterization-invariant measure with theoretical guarantees [Petzka et al., 2021, Walter et al., 2024]. While it remains an open question how these recent critiques translate to relative flatness, our findings suggest that it continues to offer a reliable signal for generalization in the settings we consider. One of the notable properties of relative flatness is that it is theoretically defined for a single, preselected layer, which enables computability independent of the model size. This formulation also allows for a non-idealized comparison with neural collapse, which is similarly defined in terms of single-layer properties, thereby enhancing the precision of our analysis. At the same time, it raises questions about how to characterize embeddings in other layers, since prior work [Zhang et al., 2022, Adilova et al., 2024, Ansuini et al., 2019] has shown that different representations and layers exhibit distinct properties, even though relative flatness appears to behave similarly across layers [Walter et al., 2025].

The theoretical link between relative flatness and generalization depends on two key assumptions: that labels are locally constant and that the features at the selected embedding layer are representative of the data distribution. In our settings (standard image classification, modular arithmetic tasks and NLP tasks) both assumptions are well justified: (i) the assumption of locally constant labels underlies the very notion of adversarial robustness, and its empirical validity is supported by our experimental findings, and (ii) we show in Appendix D that representativeness improves with the onset of generalization. Nonetheless, these assumptions may not hold in other domains such as structured prediction, high-noise regression, or complex real-world tasks, where label semantics may vary significantly under small input changes.

One of our most surprising results is that regularizing networks away from flat minima consistently induces delayed generalization, mimicking grokking even on real-world datasets and architectures. This underscores the functional importance of relative flatness, not just its correlation with generalization. It also opens new directions for controlling training dynamics via geometric regularization. However, the extent to which these artificially induced grokking phases mirror the mechanisms of standard grokking remains to be understood.

Finally, while penalizing relative flatness reliably delays generalization, encouraging flatter minima does not substantially improve it [Adilova et al., 2023a]. Even SAM, which improves training results in vision tasks, surprisingly does not actually encourage only flatter minima [Andriushchenko and Flammarion, 2022]. This asymmetry supports the long-held view that stochastic gradient descent (SGD) already biases toward flat minima [Hochreiter and Schmidhuber, 1997, Keskar et al., 2017], and that relative flatness may serve more effectively as a diagnostic and interpretive tool rather than a universal optimization target.

While our findings are robust across datasets and architectures, our conclusions are necessarily limited in scope. We focus exclusively on classification networks and generative language models trained with standard objectives and optimizers; whether our results extend to contrastive pretraining, or large-scale language models remains an open question.

We challenge the assumption that neural collapse is essential for generalization. Through grokking and targeted regularization, we isolate flatness as the necessary factor. Our theoretical and empirical results jointly establish flatness as the geometric driver of generalization.

## Acknowledgements

This research is supported by the Federal Ministry of Education and Research of Germany and the state of North Rhine-Westphalia as part of the Lamarr Institute for Machine Learning and Artificial Intelligence, and received support from the Cancer Research Center Cologne Essen (CCCE).

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

# A Proof of Proposition 5.3

First, we provide a technical lemma that establishes that, under Neural Collapse, the combined contribution of the bias and the projection of the class mean onto the global mean is identical across classes. This means that the logits are effectively governed solely by the geometric structure of the class means relative to each other.

**Lemma A.1.** *Suppose that the neural collapse conditions* $(NC1) - (NC4)$ *in the limit apply as in Section 5 (see also below). Then there is a constant $d$ such that $\lambda(\mu_j - \mu_g)^\top \mu_g + b_j = d$ for all $j$.*

*Proof.* For all $h$, the following set of equations holds:

$$
\begin{aligned}
\text{argmax}_j \ \lambda(\mu_j - \mu_g)^\top h + b_j \ &\overset{(NC3)}{=} \ \text{argmax}_j \ w_j^\top h + b_j \\
&\overset{(NC4)}{=} \ \text{argmin}_j \ \|h - \mu_j\| \\
&= \ \text{argmin}_j \ \|(h - \mu_g) - (\mu_j - \mu_g)\|^2 \\
&= \ \text{argmin}_j \ \|h - \mu_g\|^2 + \|\mu_j - \mu_g\|^2 - 2(h - \mu_g)^\top(\mu_j - \mu_g) \\
&\overset{(NC2)}{=} \ \text{argmin}_j \ \|h - \mu_g\|^2 + M^2 - 2(h - \mu_g)^\top(\mu_j - \mu_g) \\
&= \ \text{argmax}_j \ 2(\mu_j - \mu_g)^\top(h - \mu_g) - \|h - \mu_g\|^2.
\end{aligned}
$$

Let $\epsilon > 0$ be a real number and $c \in \{1, \ldots, k\}$. Since the above equations hold for all $h$, they hold in particular for all $h(c, \epsilon) = \frac{\epsilon}{M}(\mu_c - \mu_g) + \mu_g$. Note that $\|h(c, \epsilon) - \mu_g\|^2 = \epsilon^2$.

Substituting $h(c, \epsilon)$ for $h$ in the above equation and letting $d_j = \lambda(\mu_j - \mu_g)^\top \mu_g + b_j$ gives us that

$$
\begin{aligned}
\text{argmax}_j \ \frac{\lambda\epsilon}{M}(\mu_j - \mu_g)^\top(\mu_c - \mu_g) + d_j \ &= \ \text{argmax}_j \ \frac{2\epsilon}{M}(\mu_j - \mu_g)^\top(\mu_c - \mu_g) - \epsilon^2 \\
&= \ \text{argmax}_j \ (\mu_j - \mu_g)^\top(\mu_c - \mu_g) \\
&\overset{(NC2)}{=} \ c
\end{aligned}
$$

This implies that

$$
\frac{\lambda\epsilon}{M}(\mu_c - \mu_g)^\top(\mu_c - \mu_g) + d_c \geq \frac{\lambda\epsilon}{M}(\mu_j - \mu_g)^\top(\mu_c - \mu_g) + d_j, \ j \neq c
$$

Using $(NC2)$, this gives that

$$
\frac{\lambda\epsilon M k}{k - 1} + d_c \geq d_j, \ j \neq c
$$

But since $\epsilon > 0$ was arbitrary, we can let $\epsilon \to 0$ to get that $d_c \geq d_j$ for all $j \neq c$. Finally, also $c$ was arbitrarily chosen, which implies that $d_j = d_c$ for all $j, c$. Hence, $d_j = \lambda(\mu_j - \mu_g)^\top \mu_g + b_j$ is constant over $j$. $\qquad \square$

We now provide a second technical lemmma showing that relative flatness is upper bounded by the simplified version $\|w\|^2 \text{Tr}(H(w))$, for which we have a closed form expression in terms of logits.

**Lemma A.2.** *Let $H(w)$ be the Hessian wrt. the penultimate layer weights $w \in \mathbb{R}^{d \times m}$ at a minimum and $\kappa_\phi(w)$ the relative flatness measure. Then*

$$
\kappa_\phi(w) \leq \|w\|^2 \text{Tr}(H(w)) \ .
$$

*Proof.* Recall that

$$
\kappa_\phi(w) = \sum_{s,s'} \langle w_s, w_{s'} \rangle \text{Tr}(H_{s,s'}) \leq \sum_{s,s'} |\langle w_s, w_{s'} \rangle \text{Tr}(H_{s,s'})| \leq \sum_{s,s'} \|w_s\| \|w_{s'}\| \, |\text{Tr}(H_{s,s'})| \ .
$$

Since $w$ is at a minimum, $H$ is positive semi-definite (psd) and therefore $(\mathrm{Tr}(H_{s,s'}))_{s,s'}$ is also psd. Then it holds that

$$|\mathrm{Tr}(H_{s,s'})| \leq \sqrt{\mathrm{Tr}(H_{s,s})\mathrm{Tr}(H_{s',s'})} \ .$$

Summing over all $s, s'$ and using the Cauchy-Schwartz inequality we get

$$\sum_{s,s'} \|w_s\|\|w_{s'}\|\sqrt{\mathrm{Tr}(H_{s,s})\mathrm{Tr}(H_{s',s'})} \leq \left(\sum_s \|w_s\|\sqrt{\mathrm{Tr}(H_{s,s})}\right)^2 \ ,$$

and using symmetrie yields

$$\kappa_\phi(w) \leq \|w\|^2 \left(\sqrt{\mathrm{Tr}(H(w))}\right)^2 = \|w\|^2 \mathrm{Tr}(H(w)) \ .$$

$\square$

With this we can proof the theorem, which we restate together with the neural collapse criteria for convenience.

**Assumptions A.3** (Neural Collapse [Papyan et al., 2020]). Let $\phi(x)$ denote the penultimate layer feature representation of an input $x \in \mathcal{X}$, and for a dataset $D \subset \mathcal{X}$, let $D_c \subseteq D$ denote the set of inputs belonging to class $c \in \{1, \ldots, k\}$. Then the four neural collapse criteria are

(NC1) Feature representations collapse to class means: $\phi(x) = \mu_c$ for all $x \in D_c$.

(NC2) All class means $\mu_c$ have equal distance to the global mean $\mu_g$, i.e., $\|\mu_c - \mu_g\|_2 = M$, and they form a centered equiangular tight frame (ETF), so that $(\mu_j - \mu_g)^\top(\mu_c - \mu_g) = -\frac{M^2}{k-1}$ for $j \neq c$. We additionally assume that $\|\mu_c\| \leq M$ for each $c$. That is, we assume that the center of the ETF is not diverging using the same constant $M$ for simplicity.

(NC3) The classifier weights align with class means, i.e. $w_j = \lambda(\mu_j - \mu_g)$ for all $j$ and some $\lambda > 0$.

(NC4) $\mathrm{argmax}_j w_j^\top h + b_j = \mathrm{argmin}_j \|h - \mu_j\|$.

**Proposition A.4.** *Let $f(x) = softmax(w\phi(x) + b)$ be a neural network with softmax output and trained with cross-entropy loss, where $w \in \mathbb{R}^{k \times d}$ denotes the final-layer weight matrix classifying into $k$ classes and $\phi(x)$ is the penultimate-layer representation. Assume that the classical neural collapse holds in the limit as above. Then relative flatness $\kappa_\phi(w)$ is bounded by:*

$$\kappa_\phi(w) \leq \lambda^2 k^3 M^4 \cdot \frac{e^{-\lambda M^2 \cdot \frac{k}{k-1}}}{\left(1 + (k-1)e^{-\lambda M^2 \cdot \frac{k}{k-1}}\right)^2}$$

*In particular, for sufficiently large $\lambda$, this yields the asymptotic bound:*

$$\kappa_\phi(w) \lesssim \lambda^2 k^3 M^4 e^{-\lambda M^2 \cdot \frac{k}{k-1}},$$

*which decays exponentially in $\lambda$.*

*Proof.* Lemma A.1 shows that the NC conditions imply that there is a constant $d$ such that $b_j + \lambda(\mu_j - \mu_g)^\top\mu_g = d$ for all class labels $j = 1, \ldots, k$. For $x_c \in D_c$, the logits of $f$ are

$$
\begin{aligned}
w_j^\top\phi(x_c) + b_j &= \lambda(\mu_j - \mu_g)^\top\mu_c + b_j \\
&= \lambda(\mu_j - \mu_g)^\top(\mu_c - \mu_g) + \underbrace{\lambda(\mu_j - \mu_g)^\top\mu_g + b_j}_{=d} = \lambda(\mu_j - \mu_g)^\top(\mu_c - \mu_g) + d \ ,
\end{aligned}
$$

and the softmax probabilities become:

$$\hat{y}_c(x_c) = \frac{1}{1 + (k-1)e^{-\lambda\delta}}, \quad \hat{y}_j(x_c) = \frac{e^{-\lambda\delta}}{1 + (k-1)e^{-\lambda\delta}}, \quad j \neq c,$$

with margin

$$\delta := M^2 - \left(-\frac{M^2}{k-1}\right) = M^2 \cdot \frac{k}{k-1}.$$

The trace of the Hessian for input $x_c \in D_c$ is (cf. Walter et al. [2024]):

$$\text{Tr}(H(x_c, w)) = \sum_{j=1}^{k} \hat{y}_j(x_c)(1 - \hat{y}_j(x_c)) \cdot \|\phi(x_c)\|^2 = M^2 \cdot \sum_{j=1}^{k} \hat{y}_j(x_c)(1 - \hat{y}_j(x_c)).$$

$$= M^2 \frac{(k-1)e^{-\lambda\delta} \cdot (2 + (k-2)e^{-\lambda\delta})}{(1 + (k-1)e^{-\lambda\delta})^2}.$$

Thus,

$$\text{Tr}(H(w)) \leq M^2 \cdot \frac{(k-1)ke^{-\lambda\delta}}{(1 + (k-1)e^{-\lambda\delta})^2}.$$

Since $\|w\|^2 = \lambda^2 \sum_j \|\mu_j - \mu_g\|^2 = \lambda^2 k M^2$, we obtain with Lemma A.2:

$$\kappa_\phi(w) \leq \lambda^2 k M^2 \cdot M^2 \cdot \frac{(k-1)ke^{-\lambda\delta}}{(1 + (k-1)e^{-\lambda\delta})^2} \leq \lambda^2 k^3 M^4 \cdot \frac{e^{-\lambda M^2 \cdot \frac{k}{k-1}}}{\left(1 + (k-1)e^{-\lambda M^2 \cdot \frac{k}{k-1}}\right)^2}.$$

This yields the desired bound, and for large $\lambda$, the denominator tends to 1, giving the asymptotic behavior. $\square$

**Relationship between Neural Collapse and Relative Flatness.**   Neural Collapse (NC) characterizes a geometric property of the feature representations in the penultimate layer, which we quantify by the Neural Collapse Coefficient (NCC). Importantly, NCC is invariant to rescaling of the last-layer weights, and therefore independent of the scalar $\lambda$ that controls their magnitude. Consequently, NCC itself does not directly imply flatness: the loss landscape can remain sharp even in the presence of perfect collapse. However, in the NC regime, the network structure enables flatness to emerge naturally. As training with the cross-entropy loss continues, the model can reduce the training loss further by increasing $\lambda$, since the softmax probabilities satisfy $\hat{y}_c(x_c) \to 1$ and $\hat{y}_j(x_c) \to 0, j \neq c$ as $\lambda \to \infty$. This rescaling leaves the feature geometry (and hence NCC) unchanged but decreases the relative flatness measure, effectively leading to flatness without altering the collapsed representation. In summary, NC itself does not guarantee flatness, but in combination with CE loss training leads to flatness: it defines a feature geometry that allows the network to reach a flat minimum through continued optimization. Observing NC is thus a strong empirical indicator of eventual generalization, but not a necessary condition for it. Conversely, models can achieve generalization and flatness through other representational structures that do not exhibit collapse.

## B   Training Setups for NCC and Additional Experiments

We evaluate our method on the CIFAR-10 dataset using the standard training (50,000 samples) and test (10,000 samples) splits. Input images are normalized using the channel-wise mean and standard deviation of $(0.5, 0.5, 0.5)$ and $(0.5, 0.5, 0.5)$, respectively, as implemented in `transforms.Normalize`. No data augmentation is applied in the main experiments.

All models are trained using stochastic gradient descent (SGD) with a fixed learning rate of $0.01$, a batch size of $64$, and no weight decay. The regularization coefficient is set to $\lambda = 10^{-3}$. Momentum is $0.9$. Training is performed for $250$ epochs without the use of a learning rate scheduler or early stopping.

We use the ResNet-18 architecture as implemented in `torchvision` version 0.19.1, without any modifications. In particular, we retain the original $7 \times 7$ kernel size in the first convolutional layer and the following max pooling layer, despite the smaller resolution of CIFAR-10 images.

All results in the main experiments are reported as the average over three independent runs with random seeds $1$, $42$, and $15213$ to account for variability due to random initialization. Experiments are conducted using PyTorch 2.4.1 on a single NVIDIA A100 GPU with 80GB of memory. Training each model for 250 epochs takes approximately 10.5 hours.

The loss corresponding to Figure 2 is presented in Figure 5.

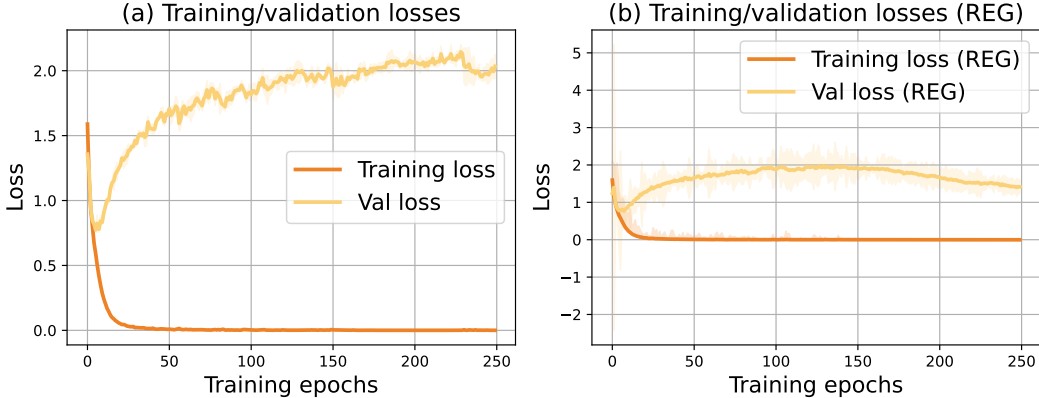

Figure 5: Results of training and validation losses in the NCC experiments.

**Effect of Different $\lambda$ Values of NCC Regularizer**    In our main experiments, we apply $\lambda = 1 \times 10^{-3}$ using three random seeds. To study the effect of different $\lambda$ values, we conduct a hyperparameter search experiment using a range of $\lambda = \{1 \times 10^{-2}, 1 \times 10^{-3}, 1 \times 10^{-4}, 1 \times 10^{-5}, 1 \times 10^{-6}\}$, evaluated with Seed 42 due to computational constraints. We have verified that results with a single seed remain consistent with the trends observed using multiple seeds.

The results are shown in Figure 6, which plots accuracy, loss, relative flatness, and neural collapse clustering (NCC) metrics. We observe that a large $\lambda$ value (e.g., $1 \times 10^{-2}$) causes training instability: although training accuracy reaches nearly 100% at the end of training, validation accuracy remains below 60%, and both training and validation accuracies collapse to around 10% around epoch 70.

Conversely, when $\lambda$ is smaller than $1 \times 10^{-3}$, the NCC regularizer has little to no effect on the training, as evidenced by negligible differences in performance metrics and NCC values compared to the baseline. When $\lambda$ is set to $1 \times 10^{-3}$, the measure of relative flatness remains low while the NCC metric increases. This supports our argument that neural collapse clustering is not necessary for generalization, as good generalization performance can still be achieved in the absence of strong NCC behavior.

**Effect of Data Augmentation in NCC**    To investigate the effect of data augmentation on NCC regularization, we apply standard transformations—*RandomCrop(32, padding=4)* and *RandomHorizontalFlip()*—and conduct experiments using $\lambda \in \{10^{-2}, 10^{-3}, 10^{-4}\}$. All inputs are normalized using per-channel means and standard deviations of $(0.4914, 0.4822, 0.4465)$ and $(0.2023, 0.1994, 0.2010)$, respectively.

We limit the range of $\lambda$ to these three values because prior results indicate that $\lambda = 10^{-4}$ has negligible effect; smaller values would therefore be redundant. Due to limited resources, we use seed 42 for this experiment, and based on consistent trends observed across seeds in other experiments, we find this sufficient for reliable comparison.

The results, shown in Figure 7, indicate that with $\lambda = 10^{-2}$, training becomes unstable and performance significantly deteriorates. With $\lambda = 10^{-3}$ and $10^{-4}$, data augmentation prevents the NCC metric from increasing as it does in the non-augmented setting—suggesting that NCC regularization is less effective when strong implicit regularization is already applied. In particular, $\lambda = 10^{-3}$, which was effective without augmentation (low measure of relative flatness and increased NCC), does not produce the same NCC increase under augmentation. At $\lambda = 10^{-4}$, the NCC metric remains low in both augmented and non-augmented cases.

In these experiments, we also adopt a ResNet-18 variant tailored for CIFAR-10, following prior work: the first convolutional layer is modified to use a $3 \times 3$ kernel, and the initial max pooling layer is removed to better suit the smaller input resolution. The model with augmentation achieves higher validation accuracy (94%) compared to the non-augmented case (78%).

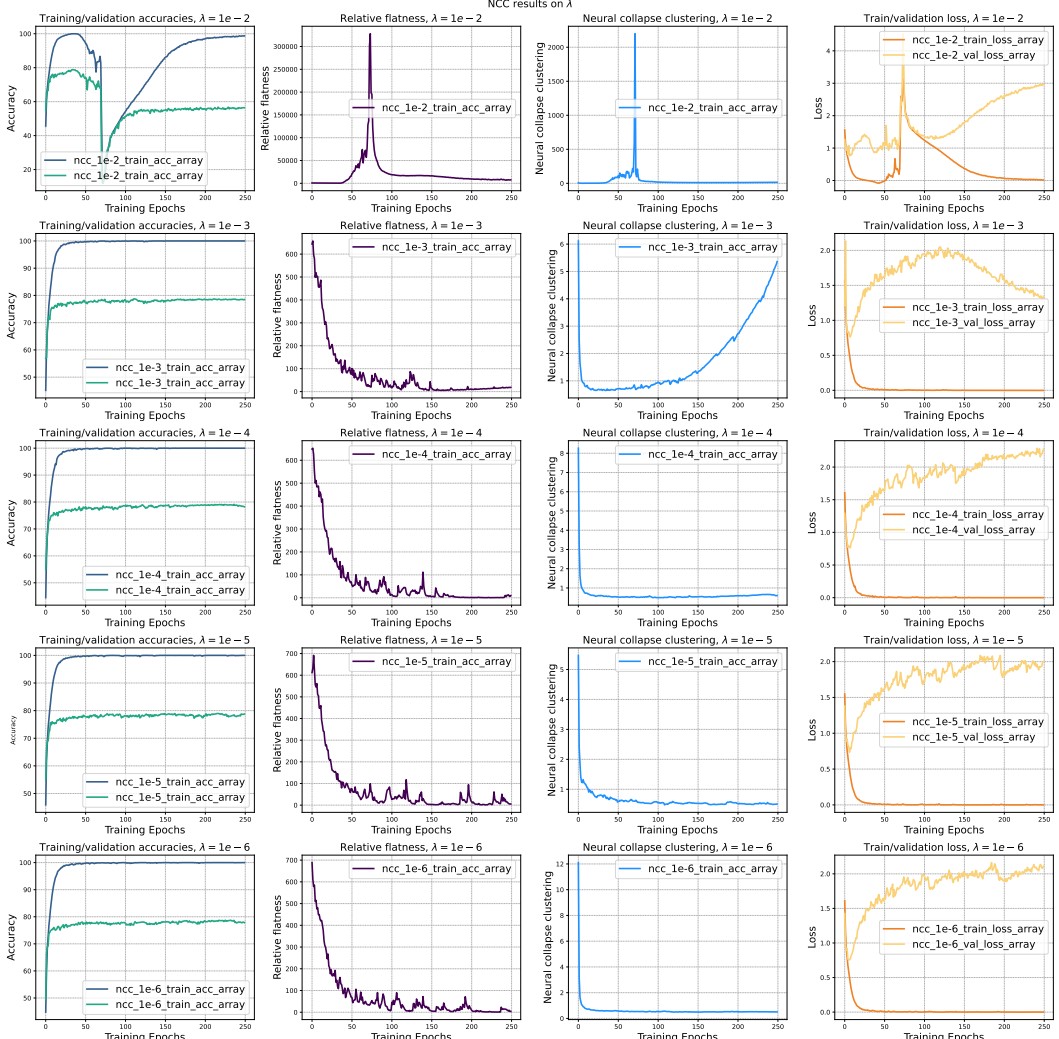

Figure 6: Results of different $\lambda$ values for the NCC regularizer. Each row contains training/validation accuracies, measure of relative flatness, NCC measurement and training/validation losses for a particular $\lambda$ value.

## C  Training Setups for Regularizing Relative Flatness and Experimental Results

### C.1  Aanalysis of Relative Flatness Regularizer and Gradients

As stated in Section 6, the loss employed to regularize relative flatness is as follows:

$$\mathcal{L}_{\text{RF\_REG}} = \mathcal{L}_{\text{CE}} - \lambda \cdot \kappa^\phi_{\text{Tr}}(\mathbf{w}).$$

Let a single training example be $(x^{(i)}, y^{(i)})$ with label $y^{(i)} \in \{1, \dots, C\}$. The penultimate representation is $\phi^{(i)} \in \mathbb{R}^d$, and the linear classifier has parameters $W \in \mathbb{R}^{C \times d}$, $b \in \mathbb{R}^C$. The logits and softmax probabilities are

$$z^{(i)} = W\phi^{(i)} + b, \qquad p_c^{(i)} = \frac{\exp(z_c^{(i)})}{\sum_{j=1}^C \exp(z_j^{(i)})}, \quad c = 1, \dots, C.$$

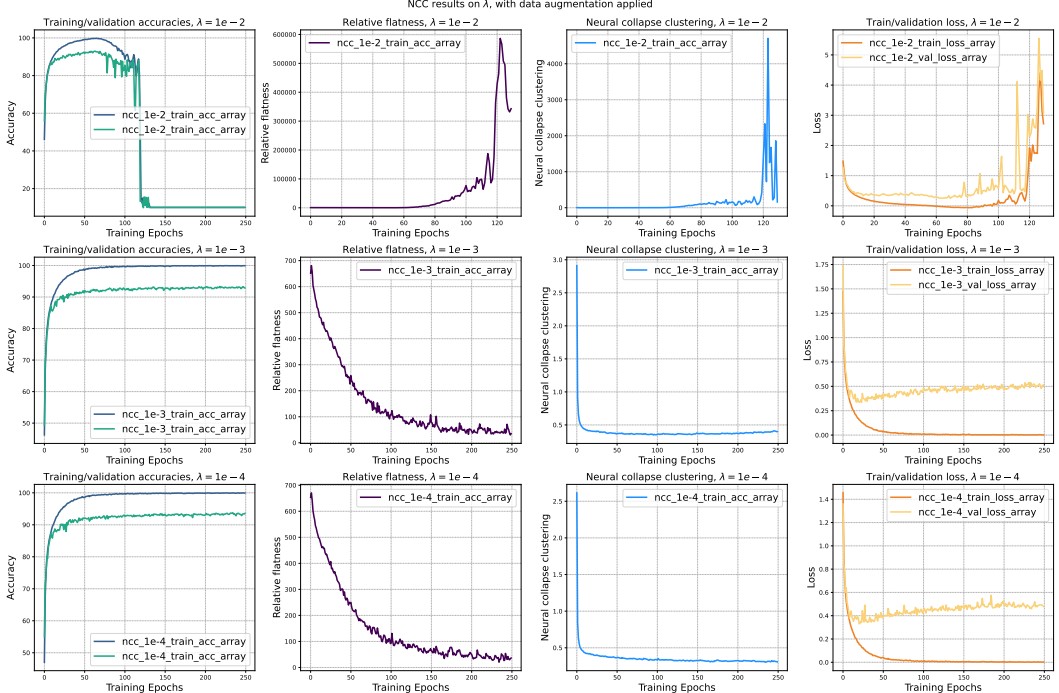

Figure 7: Results of different $\lambda$ values for the NCC regularizer when data augmentation is applied. Each row contains training/validation accuracies, measure of relative flatness, NCC measurement and training/validation losses for a particular $\lambda$ value.

The per-sample cross-entropy loss is

$$\mathcal{L}_{\mathrm{CE}}^{(i)} = -\log p_{y^{(i)}}^{(i)}.$$

Based on the derivation in Walter et al. [2024], the per-sample relative flatness with cross-entropy loss is

$$\kappa_{\mathrm{Tr}}^{\phi\,(i)}(\mathbf{w}) = \|W\|_F^2 \sum_{c=1}^{C} p_c^{(i)} \left(1 - p_c^{(i)}\right) \|\phi^{(i)}\|_2^2,$$

and the total per-sample loss is:

$$\mathcal{L}_{\mathrm{RF\_REG}}^{(i)} = -\log p_{y^{(i)}}^{(i)} - \lambda \|W\|_F^2 \left(\sum_{c=1}^{C} p_c^{(i)} \left(1 - p_c^{(i)}\right)\right) \|\phi^{(i)}\|_2^2 \tag{1}$$

$$= -\log p_{y^{(i)}}^{(i)} - \lambda \|W\|_F^2 \left(\sum_{c=1}^{C} p_c^{(i)} - \sum_{c=1}^{C} \left(p_c^{(i)}\right)^2\right) \|\phi^{(i)}\|_2^2 \tag{2}$$

$$= -\log p_{y^{(i)}}^{(i)} - \lambda \|W\|_F^2 \left(1 - \|p^{(i)}\|_2^2\right) \|\phi^{(i)}\|_2^2. \tag{3}$$

For simplicity, let $U(p^{(i)}) = 1 - \|p^{(i)}\|_2^2$, where $0 \le U(p^{(i)}) \le 1 - \frac{1}{C}$. Then $R^{(i)} = \|W\|_F^2 \, U(p^{(i)}) \, \|\phi^{(i)}\|_2^2$. The per-sample loss can be re-written as

$$\mathcal{L}_{\mathrm{RF\_REG}}^{(i)} = -\log p_{y^{(i)}}^{(i)} - \lambda R^{(i)}.$$

To encourage sharper minima, the optimizer tends to increase $\|W\|$ and $\|\phi\|$ under high uncertainty $U(p^{(i)})$. On the contrary, the cross-entropy loss aims to minimize prediction uncertainty, which conflicts with the objective of the regularizer. When $\lambda$ is set to a small value, the cross-entropy loss

quickly dominates the training, driving uncertainty down and effectively nullifying the influence of the regularizer. However, when $\lambda$ is large, this effect can dominate, causing parameter growth that eventually destabilizes training. To better understand this, we first look at the gradients with respect to the regularizer.

**Gradients.**   Write $S = \|\phi^{(i)}\|_2^2$ and $U = 1 - \|p^{(i)}\|_2^2$. Then

$$\frac{\partial R^{(i)}}{\partial W} = 2\,U\,S\,W \; + \; \|W\|_F^2\,S\left(\frac{\partial U}{\partial z}\right)\phi^{(i)\top},$$

$$\frac{\partial R^{(i)}}{\partial \phi} = \|W\|_F^2\left[\,S\,W^\top\left(\frac{\partial U}{\partial z}\right)^\top \; + \; 2U\,\phi^{(i)}\,\right].$$

Thus the overall gradients are

$$\frac{\partial \mathcal{L}^{(i)}}{\partial W} = \frac{\partial \mathcal{L}_{\mathrm{CE}}^{(i)}}{\partial W} - \lambda\,\frac{\partial R^{(i)}}{\partial W}, \qquad \frac{\partial \mathcal{L}^{(i)}}{\partial \phi} = \frac{\partial \mathcal{L}_{\mathrm{CE}}^{(i)}}{\partial \phi} - \lambda\,\frac{\partial R^{(i)}}{\partial \phi}.$$

Without constraints, the multiplicative term $2USW$ amplifies $\|W\|$ whenever predictions are uncertain ($U$ large) and features have large norm ($S$ large), which may cause runaway weight growth and unstable training.

$$W_{t+1} = W_t - \eta\,\left.\frac{\partial \mathcal{L}_{\mathrm{CE}}^{(i)}}{\partial W}\right|_t + \eta\lambda\left(2U_t S_t\,W_t + \|W_t\|_F^2 S_t\left(\frac{\partial U}{\partial z}\right)_t \phi_t^\top\right),$$

$$\phi_{t+1} = \phi_t - \eta\,\left.\frac{\partial \mathcal{L}_{\mathrm{CE}}^{(i)}}{\partial \phi}\right|_t + \eta\lambda\|W_t\|_F^2\left(S_t\,W_t^\top\left(\frac{\partial U}{\partial z}\right)_t^\top + 2U_t\,\phi_t\right),$$

$$b_{t+1} = b_t + \eta\lambda\|W_t\|_F^2 S_t\left(\frac{\partial U}{\partial z}\right)_t.$$

**Multiplicative growth on the classifier.**   The $W$-update includes the term $2\eta\lambda U_t\|\phi_t\|_2^2 W_t$, which multiplies $W_t$ directly and can drive growth of $\|W_t\|$ when $U_t$ and $\|\phi_t\|$ are not small. This motivates stabilization strategies.

**Stabilization by feature normalization and weight capping.**   If $\|\phi_t\|_2 = 1$ is enforced (e.g., replacing $\phi \leftarrow \phi/\|\phi\|$ in the forward pass), then $S = \|\phi_t\|_2^2 = 1$ and the amplifying factors involving $\|\phi_t\|$ vanish. The multiplicative $W$-term reduces to $2\eta\lambda U_t W_t$ (still present but weaker), and the $\phi$-update no longer scales with $\|\phi_t\|$. In particular, normalizing $\phi$ fixes $S = \|\phi\|_2^2 = 1$, eliminating the quadratic amplification in $\phi$ that otherwise accelerates the multiplicative growth of $\|W\|$.

To further prevent runaway growth of $\|W_t\|$, we apply a Frobenius norm cap after each optimizer step:
$$W_{t+1} \; \leftarrow \; \Pi_{\mathcal{B}_F}(W_{t+1}), \qquad \mathcal{B}_F = \{\,W : \|W\|_F \le B_F\,\}.$$
This projection makes the update non-expansive in the Frobenius norm and uniformly bounds all regularizer-induced gradient terms, since they scale with $\|W_t\|$ and $\|W_t\|^2$. Together, these techniques ensure that training under the relative flatness regularizer remains stable and well-conditioned.

## C.2   CIFAR-10 & Resnet-18 Training Setups and Additional Experiments

We follow the same training setup described in Section B, with two modifications: (1) the total number of training epochs is increased to $400$; and (2) the coefficient of the relative-flatness regularizer is set to 0.01. During regularized training, the weight capping value is fixed at $50$ and the penultimate representation is normalized ($\|\phi\| = 1$). We introduce temperature parameter $\tau = 2$ in the softmax when computing uncertainty $U(p^{(i)})$, so as to inflate the uncertainty signal and maintain the influence of the regularizer during training. When the regularizer is removed at Epoch 200, weight capping

remains active, but normalization of the penultimate layer is deactivated. The same procedure for weight capping and representation normalization in unplug experiments is applied to the other tasks as well. We also verify that setting the weight capping to $50$ and normalizing the penultimate representation does not affect training without the regularizer.

The training and validation loss curves associated with Figure 4 are shown in Figure 8, and the corresponding relative-flatness measurements are reported in Figure 9.

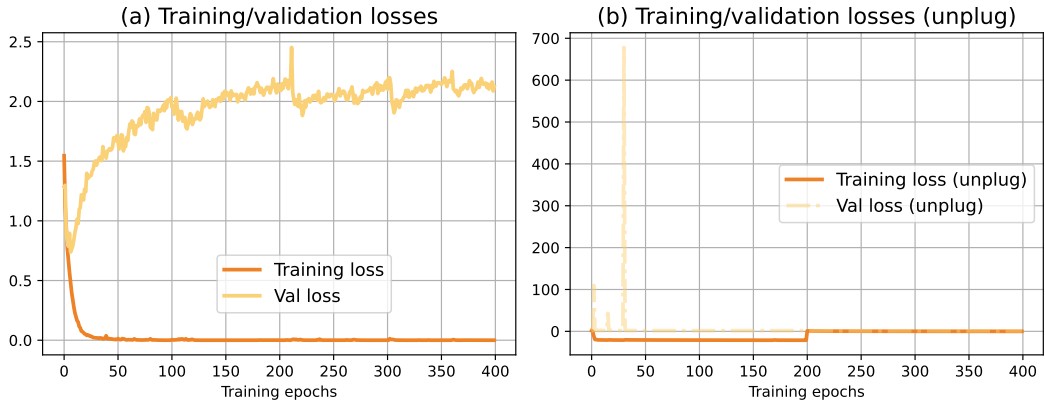

Figure 8: Results of training loss and validation loss of CIFAR-10 in the delayed generalization.

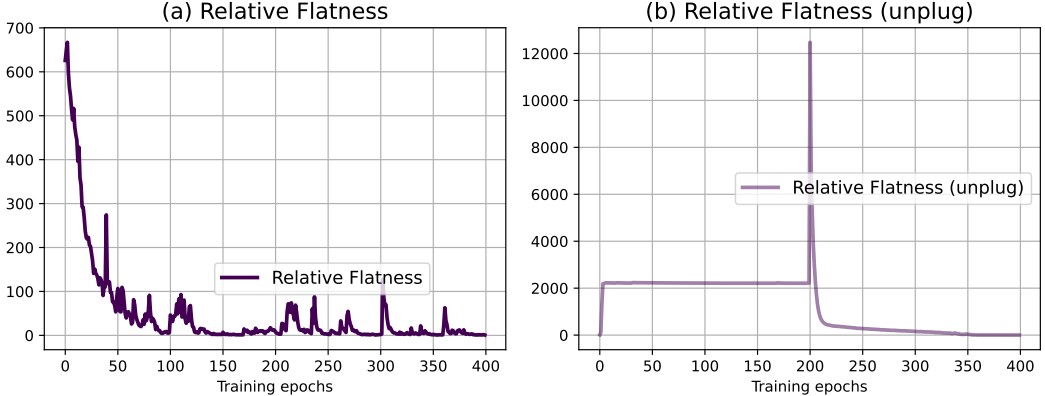

Figure 9: Results of relative flatness of CIFAR-10 in the delayed generalization.

The main experiments are initialized with seed $42$, while additional results with seeds $1$ and $15213$ are shown in Figure 10. These results demonstrate that initialization with different random seeds can drive training toward different sharp minima. Consequently, averaging across seeds does not provide meaningful insights.

In Figure 10, standard training with ResNet-18 converges reliably: training accuracy reaches nearly $100\%$, while validation accuracy stabilizes around $78\%$. The relative flatness measure decreases steadily, indicating convergence toward flatter minima. With the regularizer, training accuracy still approaches $100\%$, but validation accuracy drops to about $60\%$, well below the baseline. At the same time, the relative flatness remains high (around $2000$) and does not decrease, showing that the model is pushed into sharper solutions. These behaviors are consistent across seeds. The regularizer is removed at Epoch 200. After unplugging, the total training loss becomes positive small while the model continues to train stably. To encourage escape from the sharp minima, we reset the learning rate to $1 \times 10^{-1}$, apply a cosine annealing schedule, and introduce a weight decay of $2 \times 10^{-4}$. As shown in Figure 4, validation accuracy rises toward $78\%$, aligning with the standard training baseline. The relative flatness (Figure 9) also decreases and becomes indistinguishable from the

baseline, confirming that unplugging enables the model to recover from the sharp minima induced by the regularizer.

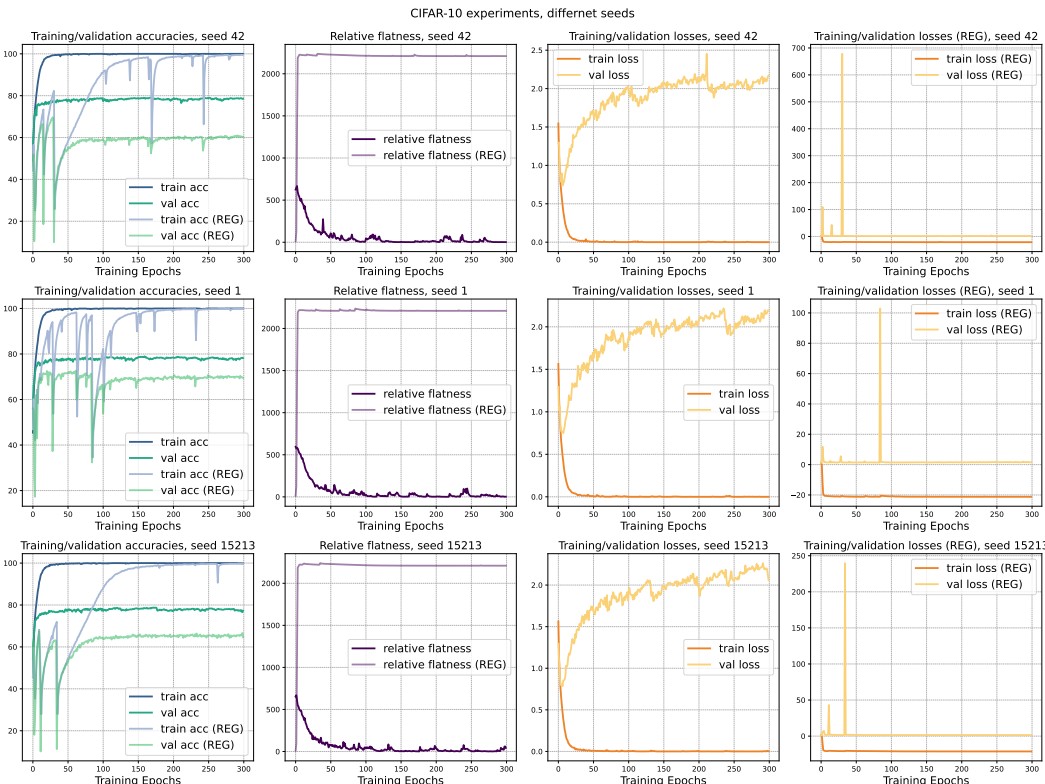

Figure 10: Resnet18 training results with different random seeds (42, 1, 15213), comparing standard fine-tuning and regularized training (REG).

**Effect of $\lambda$ Values of Relative Flatness Regularizer.** To investigate the effect of regularization strength, we conduct experiments with coefficients $\lambda \in \{0.5, 0.1, 10^{-3}, 10^{-4}\}$, while keeping all other settings (model architecture, training procedure, and random seed 42) consistent with the main experiments, except the number of training epochs is 300. Results are shown in Figure 11.

For large regularization strengths ($\lambda = 0.5$ and $0.1$), training crashes, causing both training and validation accuracies to remain low. For a moderate value ($\lambda = 10^{-3}$), training accuracy reaches nearly 100%, and validation accuracy remains slightly below the 78% baseline. Although the relative flatness stays high, the near-optimal validation accuracy indicates that an intermediate $\lambda$ can drive the network toward sharp minima without significantly degrading performance. For a smaller value ($\lambda = 10^{-4}$), training remains stable across 300 epochs. The relative flatness decreases rapidly to a low value, the training loss converges smoothly to zero, and the validation loss stabilizes. However, when $\lambda$ is set too small (below $10^{-4}$), the influence of the regularizer essentially disappears, and the model behaves like the baseline model without any affecting from relative-flatness regularization.

**Effect of Weight Cap Values in the Relative-Flatness Regularizer.** We examine the effect of different weight cap values (WC $\in \{20, 100, 150, 200\}$) while keeping all other settings identical to the main experiments. Results are shown in Figure 12.

With a small cap (WC $= 20$), training remains stable: the relative flatness is still high but stays below 2000, the level observed in the main experiment, and validation accuracy reaches about 75%, only 3% lower than the baseline. At WC $= 100$, training appears superficially stable within 300 epochs, as validation accuracy plateaus around 50% and training accuracy shows potential to approach 100% with extended training. However, the relative flatness increases rapidly to nearly four times the level reported in Figure 10, raising concerns about whether training can safely and reliably reach full

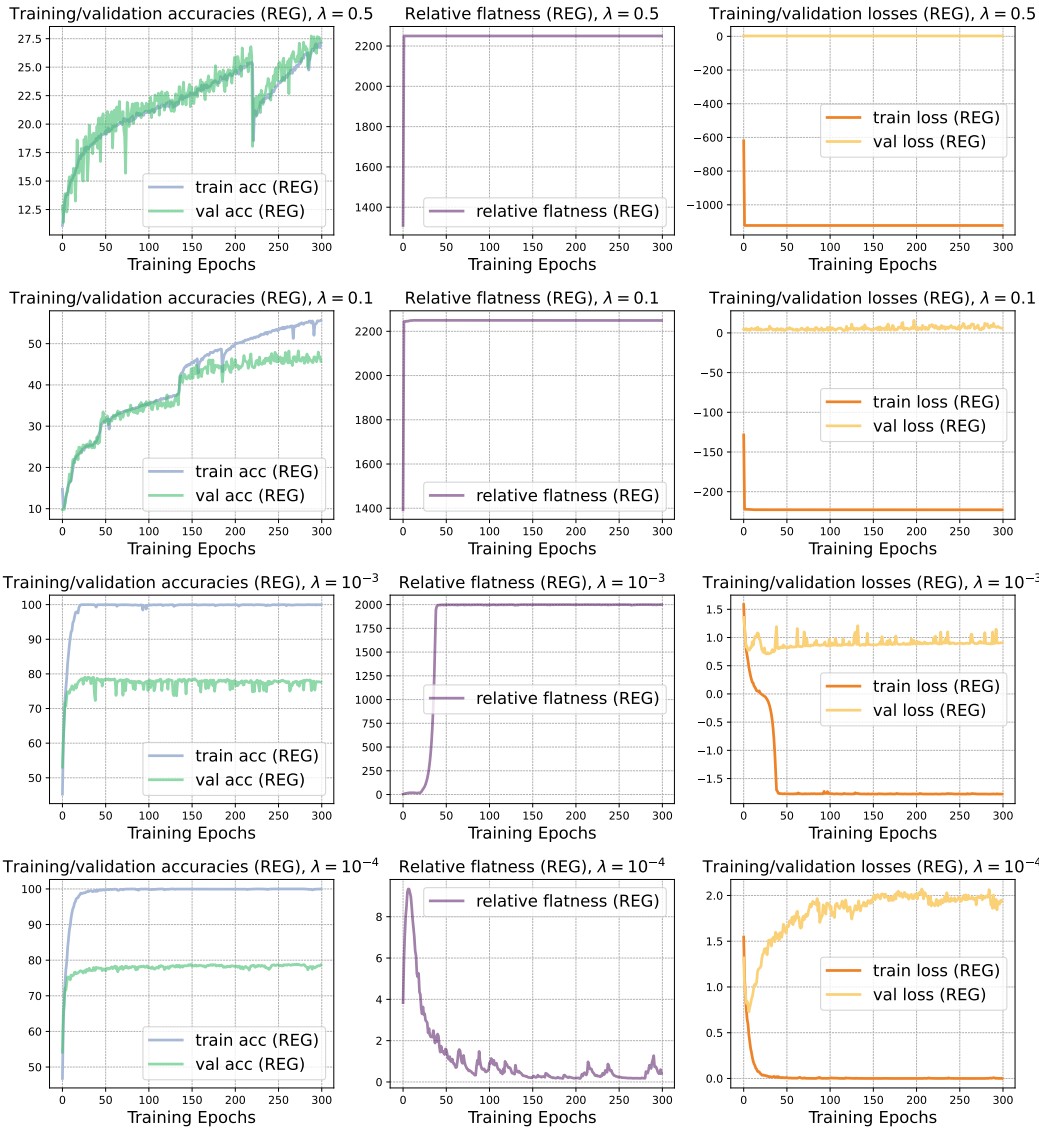

Figure 11: Results of various $\lambda$ values on CIFAR-10.

convergence. With larger caps (WC = 150 and 200), training collapses entirely: both training and validation accuracies remain at very low levels, the relative flatness diverges to extremely large values (above $10^5$), and the losses oscillate with high variance, indicating a complete failure to converge.

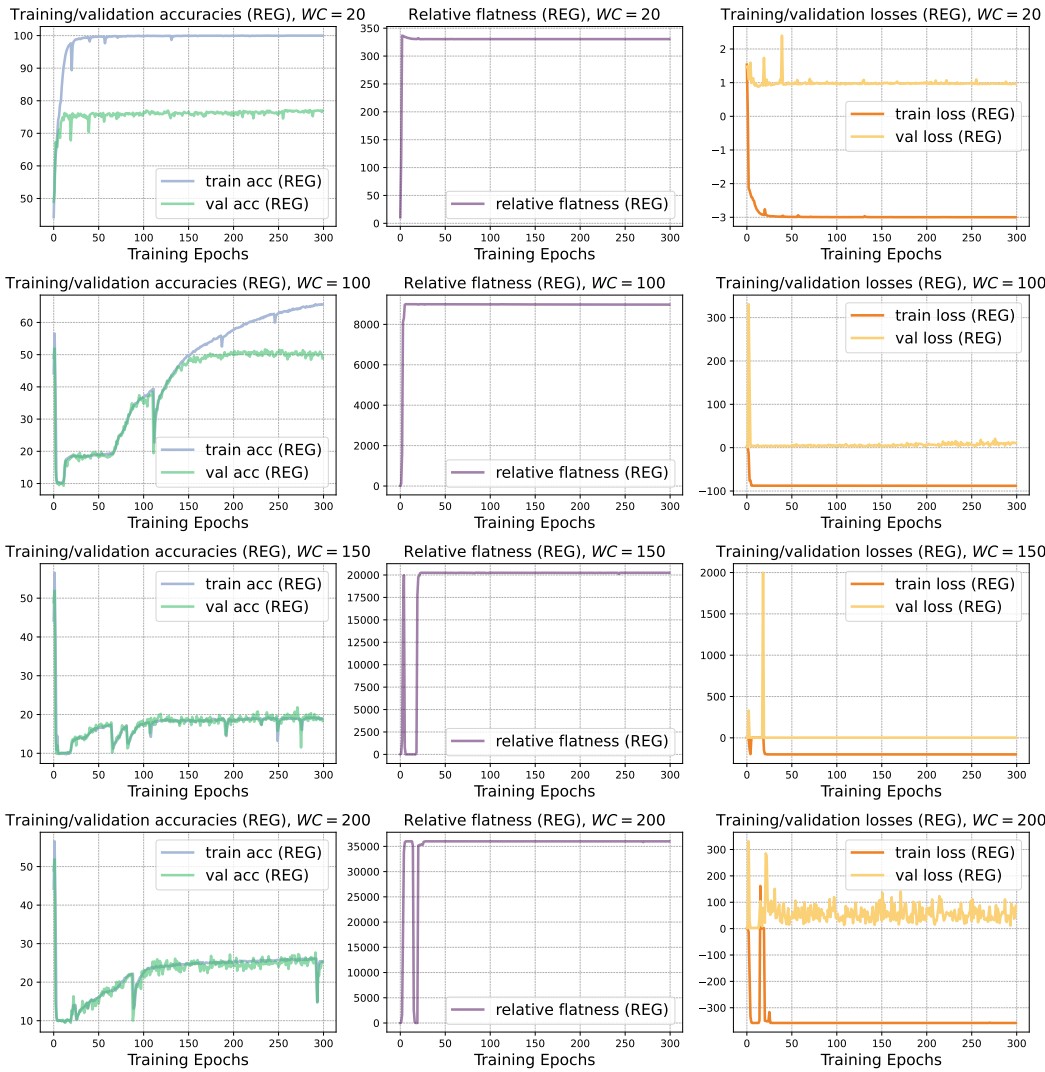

Figure 12: Results of various values of weight capping on CIFAR-10.

## C.3 Imagenet-100 & ViT Training Setups and Experimental Results

We evaluate the relative-flatness regularizer on the ImageNet-100 dataset using the standard train/test split. Input images are normalized with channel-wise means of [0.485, 0.456, 0.406] and standard deviations of [0.229, 0.224, 0.225] using `transforms.Normalize`. No data augmentation is applied in the main experiments.

We train a vision transformer (ViT-tiny) model from scratch with *all dropout components* disabled. This design excludes the generalization effects introduced by dropout. Dropout encourages smoother solutions and better generalization, whereas the relative-flatness regularizer drives the model toward sharper minima. Combining them would create conflicting objectives. The coefficient of the regularizer is set to $\lambda = 10^{-2}$. To induce delayed generalization, the regularizer is removed after epoch 150. We optimize with SGD using a fixed learning rate of 0.01, momentum of 0.9, no weight decay, and no Nesterov acceleration. The temperature $\tau$ is set to 2 when computing the uncertainty of the regularizer, and the weight capping is set to $150$. Training runs for 300 epochs without learning rate scheduling or early stopping. The batch size is 256, and experiments are conducted on a single NVIDIA A100 GPU with 80 GB of memory.

The corresponding ImageNet-100 loss curve associated with Figure 4 is shown in Figure 13, and the relative flatness measurement is shown in Figure 14. Figure 15 presents the results under three random seeds (42, 1, 15213), comparing normal training with regularized training (REG).

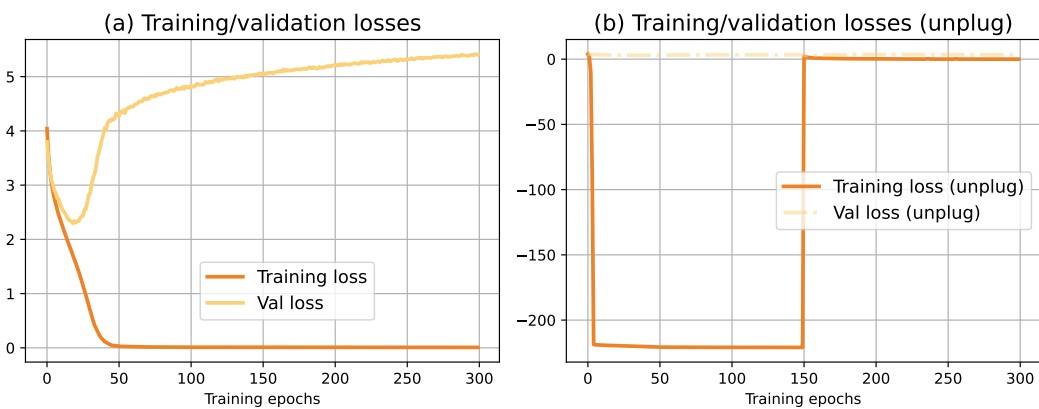

Figure 13: Results of training loss and validation loss of Imagenet-100 in the delayed generalization.

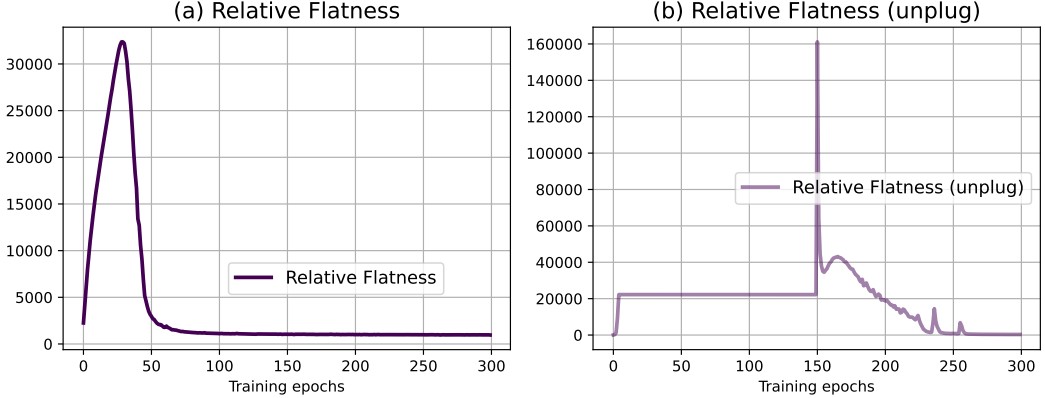

Figure 14: Results of relative flatness of Imagenet-100 in the delayed generalization.

In Figure 15, standard training converges reliably: training accuracy reaches nearly $100\%$, while validation accuracy stabilizes around $42\%$. The relative flatness measure decreases steadily, indicating convergence toward flat minima. With the regularizer, training accuracy still approaches $100\%$, but validation accuracy drops to about $38\%$, below the baseline. At the same time, the relative flatness remains very large (above 20,000) and does not decrease, showing that the model is pushed into sharp solutions. These behaviors are consistent across seeds. The regularizer is removed at Epoch 150. After unplugging, the total training loss becomes small but positive while the model continues to train stably. To encourage escape from the sharp minima, we reset the learning rate to $5 \times 10^{-2}$, apply a cosine annealing schedule, and introduce a weight decay of $2 \times 10^{-4}$. As shown in Figure 4, validation accuracy rises toward $42\%$, aligning with the standard training baseline. The relative flatness (Figure 14) also decreases and becomes indistinguishable from the baseline, confirming that unplugging helps the model recover from the sharp minima induced by the regularizer.

## C.4 SST-5 & Pretrained Language Models Training Setups and Experimental Results

We evaluate the proposed relative-flatness regularizer on fine-tuning pretrained language models TinyBERT [Jiao et al., 2020] and DistilGPT2 [Sanh et al., 2019] on the SST-5 dataset. The training hyperparameters are listed in Table 1. In these experiments, *all dropout components* in both models

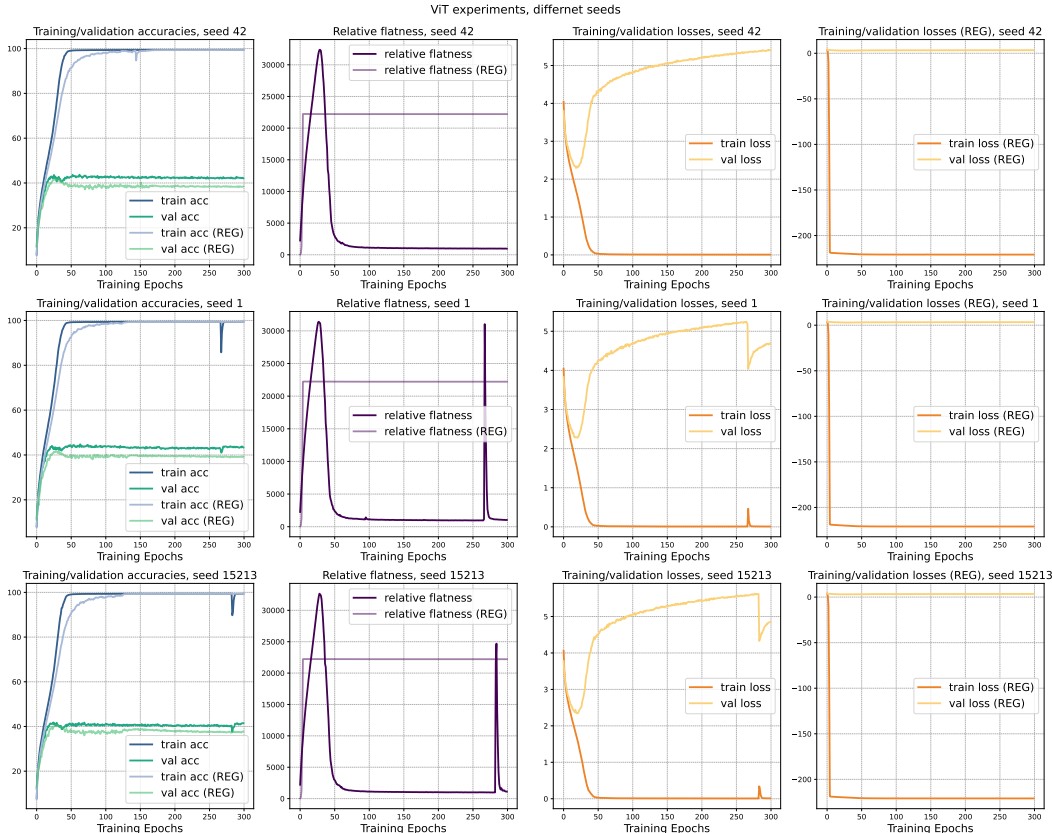

Figure 15: ViT training results with different random seeds (42, 1, 15213), comparing standard fine-tuning and regularized training (REG)

(including attention, feedforward, embedding, and classifier head dropouts) are explicitly disabled. Optimization is performed using stochastic gradient descent (SGD) with a fixed learning rate of $10^{-3}$, momentum of 0.9, no weight decay, and no Nesterov acceleration. Training runs for 300 epochs without learning rate scheduling or early stopping. Figures 16 and 18 report the results under three random seeds (42, 1, 15213), comparing standard fine-tuning with regularized training (REG). Similar to the results on CIFAR-10 and ImageNet-100, training converges to slightly different minima under different random seeds, which is reflected by minor differences among validation accuracies. Figures 17 and 19 present the unplug results with random seed 42.

Table 1: Hyperparameter settings for SST-5 fine-tuning experiments. All dropout components (attention, feedforward, embedding, and classifier head) are disabled to isolate the effect of the relative-flatness regularizer.

| Hyperparameter | TinyBERT | DistilGPT2 |
| --- | --- | --- |
| Learning rate | $10^{-3}$ | $10^{-3}$ |
| Batch size | 32 | 64 |
| Epochs | 300 | 300 |
| Random seeds | $\{42, 1, 15213\}$ | $\{42, 1, 15213\}$ |
| Weight capping | 30 | 30 |
| Regularization strength $\lambda$ | $3 \times 10^{-1}$ | $3 \times 10^{-2}$ |
| Temperature $\tau$ | 2.0 | 2.0 |
| Optimizer | SGD | SGD |

For TinyBERT (Figure 16), standard fine-tuning converges reliably: the training accuracy approaches 100%, and the validation accuracy stabilizes around 44%. The relative flatness measure decreases steadily, indicating convergence to relatively flat minima. In contrast, with the regularizer, training accuracy still reaches 100%, but validation accuracy drops to about 38%, below the 44% baseline. The relative flatness measure remains high (above 700) rather than decreasing, showing that the model is driven to sharp solutions by the regularizer. These patterns are consistent across seeds.

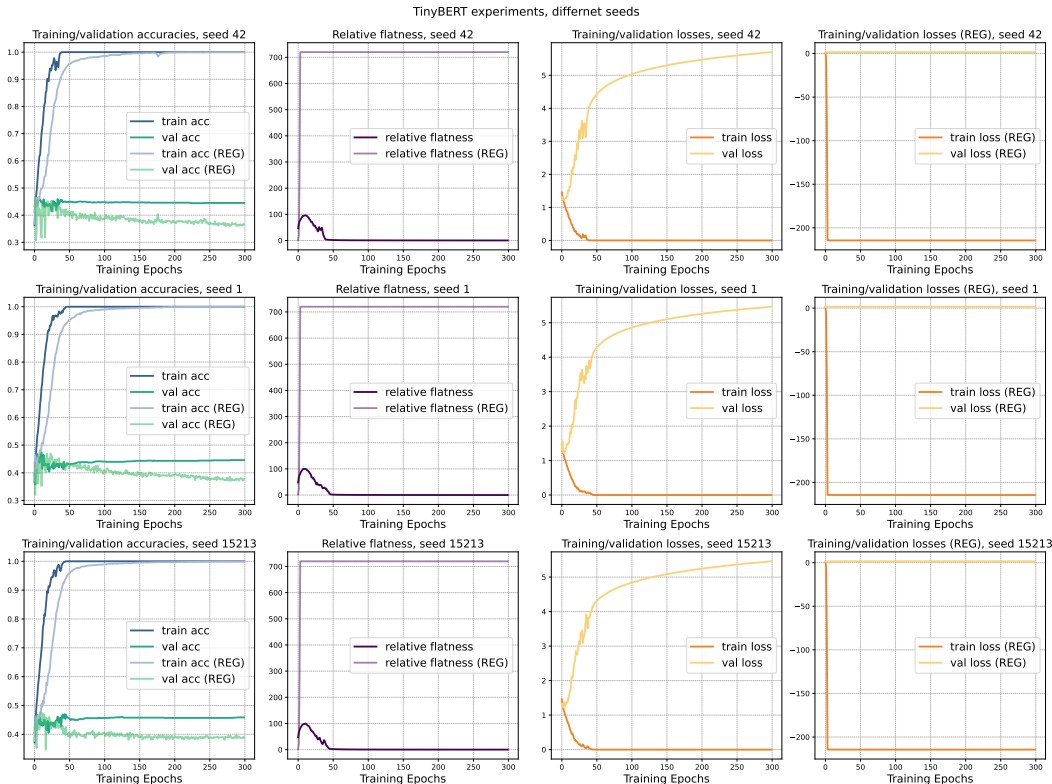

Figure 16: TinyBERT training results with different random seeds (42, 1, 15213), comparing standard fine-tuning and regularized training (REG)

For the unplug experiment in Figure 17, the regularizer is removed at Epoch 150. After unplugging, the total training loss is positively small while training remains stable in the sharp minima. To encourage the network to escape from this sharp minima, the learning rate is reset to $3 \times 10^{-3}$, a cosine annealing scheduler is applied, and a weight decay of $2 \times 10^{-4}$ is introduced. As shown, the validation accuracy rises to around 44%, close to the standard fine-tuning result. Similarly, the relative flatness decreases and becomes indistinguishable from the baseline case.

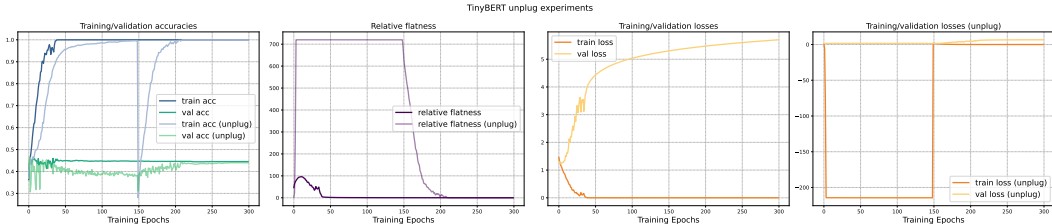

Figure 17: TinyBERT unplug experiment with random seed 42, showing the effect of removing the regularizer during training.

For DistilGPT2 (Figure 18), standard fine-tuning shows stable convergence: training accuracy reaches close to $100\%$, while validation accuracy levels off around $45\%$. The relative flatness decreases steadily during training, indicating convergence toward flatter minima. With the regularizer, training accuracy still achieves $100\%$, but validation accuracy drops to about $37\%$, below the $45\%$ baseline. In this case, the relative flatness remains high (above 600) and does not decrease, which indicates that the model is pushed into sharper solutions. These outcomes are consistent across all three random seeds.

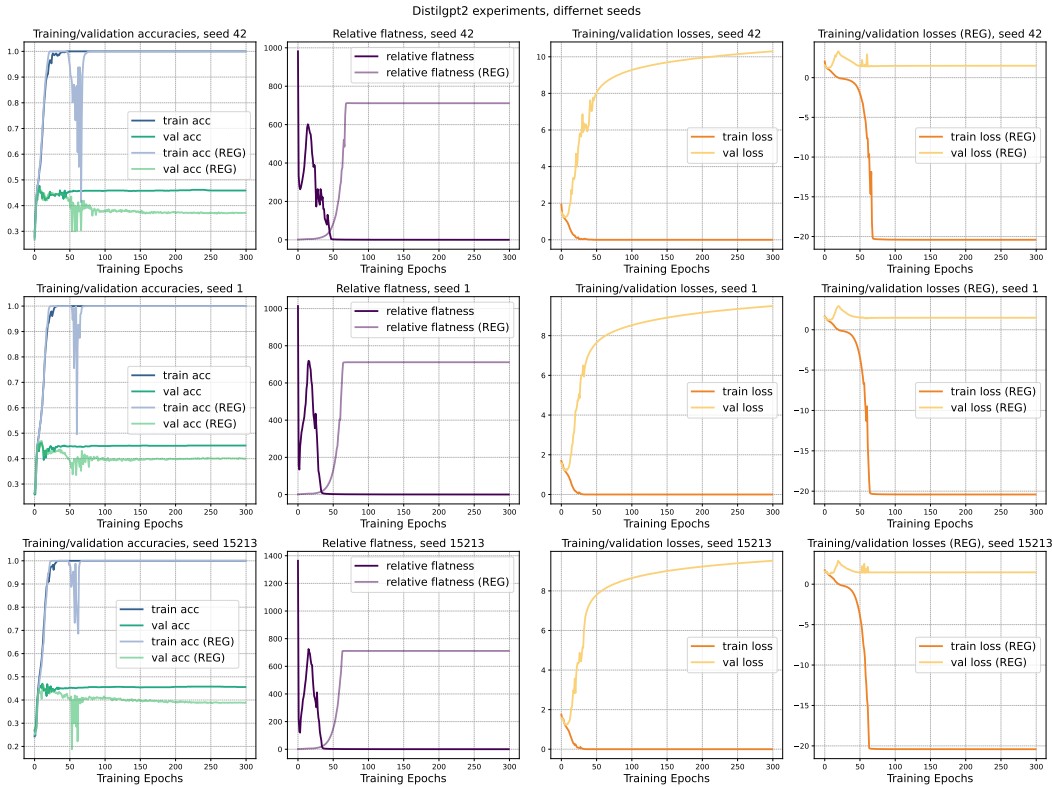

Figure 18: DistilGPT2 training results with different random seeds (42, 1, 15213), comparing standard fine-tuning and regularized training (REG).

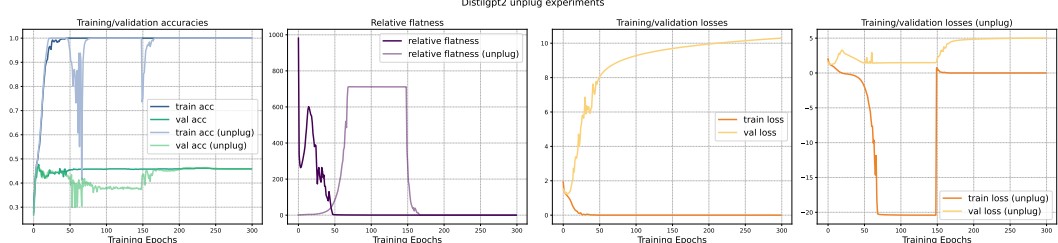

Figure 19: DistilGPT2 unplug experiment with random seed 42, showing the effect of removing the regularizer during training.

For the unplug experiment in Figure 19, the regularizer is removed at Epoch 150. Training continues smoothly, and the total training loss remains low as a positive value even in the sharp minima. To promote recovery toward flatter minima, we reset the learning rate to $5 \times 10^{-3}$, apply a cosine annealing schedule, and add a weight decay of $1 \times 10^{-4}$. After this adjustment, validation accuracy rises to about $45\%$, nearly the same as the standard fine-tuning result. The relative flatness also

decreases quickly and becomes indistinguishable from the baseline case, showing that unplugging allows the model to escape the sharp minima induced by the regularizer.

# D    Representativeness

As described in Petzka et al. [2021], representativeness is computed based on the representations from the penultimate layer of the network. Following their procedure, we use kernel density estimation (KDE) to measure representativeness (for further details, please refer to the original paper).

We employ a Gaussian kernel with a bandwidth of 1.0, and assign each training sample a weight of 0.02, chosen empirically. Consistent with the setup of the main experiments (Figure 1), all computations are averaged over three random seeds. The results are presented in Figure 20.

As shown in the results, representativeness decreases to zero when the validation accuracy reaches 100%. Moreover, we observe that improvements in representativeness coincide with the initial rise in validation accuracy during training, providing empirical support for our theoretical findings and arguments.

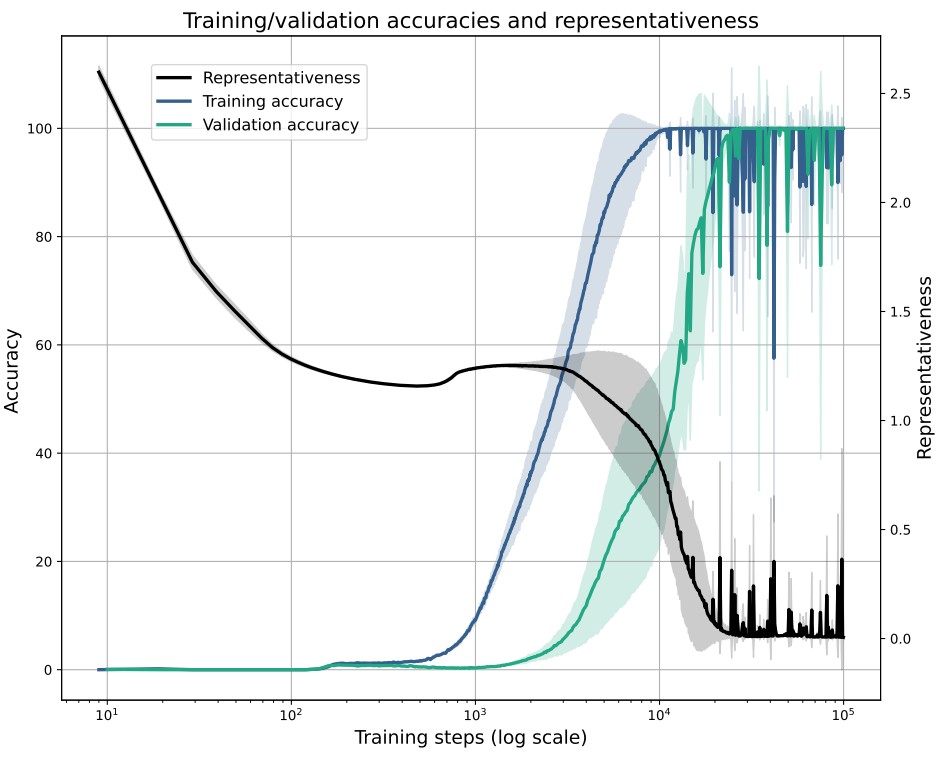

Figure 20: Results of representativenesss and training/validation accuracies in grokking.

