# OpenReview forum: "Flatness is Necessary, Neural Collapse is Not: Rethinking Generalization via Grokking"
_NeurIPS.cc/2025/Conference — NeurIPS 2025 poster_

### Official Review · Reviewer_6q1E · 2025-06-30

**Clarity:** 2
**Significance:** 2
**Originality:** 3
**Rating:** 4
**Confidence:** 3

**Summary:**

The paper studies NN generalization through the lens of neural collapse, grokking, and landscape flatness perspective. While all three concepts were previously observed to have links to the model's generalization, the paper offers a novel perspective on how these three concepts are related to each other and can potentially be used to diagnose the model's generalization. The work highlights that these concepts should rather be treated as diagnostic tools and not directly as quantities one would like to maximize in order to increase the model's performance.

**Questions:**

1. Why does the validation loss grow in almost all of your experiments? I'm referring to the Figures presented in the Appendix.

2. Why did the authors use the ResNet-18 architecture without the recommended modifications required for CIFAR-10? Without these modifications, the model achieves poor performance due to pathologically low image dimensionality across the layers when trained on CIFAR-10 (32x32), and these experiments are not representative of any sensible setting.

3. How to interpret the relative sharpness measure? What values are fine (we expect strong generalization), what values are not (the model should not generalize)? The explanation (last paragraph from the 6th section) does a poor job, and the Figures from the appendix do not provide any clarifying information. The scale is different across the plots, and it looks like regularizing **against the flatness** does not really harm this measure or maybe it's just a scale and it's hard to observe the difference?

**Ethical Concerns:**

["NO or VERY MINOR ethics concerns only"]

**Final Justification:**

As written earlier, I decied to keep my initial score, however I do akcnowledge the fact that authors clarified several points that could lead to misunderstanding, thus I rating should be treated as neutral (I'm ok with rejecting the work, e.g. for the reasons I've mentioned but at the same time, the work brings some interesting observations to the community and accepting this paper could also be ok).

**Limitations:**

Yes

**Paper Formatting Concerns:**

Line 345 -- "both empirical evidence and consistent experimental outcomes across different tasks." I find this sentence confusing. What's the difference between empirical evidence and experiment outcomes?

Line 957 - 959 -- the sentence is repeated twice, referring to two different Figures.

**Quality:**

2

**Strengths And Weaknesses:**

## Strengths
1. The whole work focuses on a single, well-defined topic. Given that the text is clearly written and structured, it's an easy-to-follow argument.
2. Generalization is one of the most fundamental concepts of ML; thus, the motivation to better understand its roots or limitations is always well justified.
3. The idea to combine seemingly related concepts (Neural Collapse, Grokking, Flatness), which are usually studied in a separate way, is an interesting approach worth studying. I'd like to point out that an interesting work with a similar approach that would be worth citing has been published at the recent ICML conference: https://arxiv.org/abs/2502.21009

## Weaknesses
1. ### Insufficient experiments on grokking, NC, and flatness
While the paper's claim to "disentangle the [Neural Collapse and flatness] contributions to the model performance by leveraging the grokking setting" is an interesting idea, the only experiment that combines these three concepts is presented in Figure 1 and performed on a modular arithmetic task. I don't fully agree with the authors that Figure 4 represents a "delayed generalization" setting, as initially, both training and validation accuracy grow. Thus, while interesting, I'm not convinced that the hypothesis about grokking/NC/flatness relationship is supported by sufficient experiments.
2. ### Imprecise claims
a) From the very beginning, the work claims the connection between NC and other phenomena. While authors note that NC consists of 4 different conditions (Section 3), in reality, the authors always check a simplified NCC condition (Definition 3.1) and argue that "NCC measure correlates with the angular separation of means, **thus capturing all the characteristics of NC**". First, the correlation is again evaluated only on a single, rather simple experiment (CIFAR-10, ResNet-18), which is not sufficient in my opinion. Second, I don't see how NCC + angular separation imply **all the characteristics of NC**, could the authors please elaborate on this? If that's true, it needs a justification; otherwise, I ask the authors to ease the tone on the connection of NC, as this is not a fully supported claim.
b) As written above, I do not fully agree with the author's statement that the experiments presented in Figure 4 present the grokking phenomenon, which is related to "delay generalization". In these experiments, the generalization occurs at the beginning, but due to the regularization, it is diminished.
3. ### Neural Collapse is not Necessary (Section 5)
I'm not sure about the rationale of this experiment. I haven't found any literature suggesting that Neural Collapse is required to achieve generalization. In fact, NC appears only after TPT (Terminal Phase of Training) on restricted setups with no augmentations and prolonged training, and we know that these factors are not necessary to achieve strong generalization for the models, so I'm confused about what the point of this argument is.

---

> ### Author Rebuttal · Authors · 2025-07-31
>
> Dear Reviewer 6qE1,
>
> Thank you for pointing out the relevance of our work and the clarity of our scope. We will address your concerns individually.
>
> **W1:** We appreciate the reviewer’s feedback and the opportunity to clarify both our use of terminology and the experimental scope. Grokking so far has been observed in very specific settings [1] which we evaluate in Fig. 1. In our Fig. 4, we observe a delayed generalization similar to grokking. We understand the reviewer’s hesitation regarding our use of the term “delayed generalization” in Figure 4, as validation accuracy does exhibit a modest initial rise. Our intent was not to claim canonical grokking behavior in this case, but rather to highlight that explicit regularization of flatness creates a pronounced delay in generalization, even in settings where grokking is not typically observed. To better reflect this distinction, we now refer to this as “grokking-like” or “induced delayed generalization” and have softened claims where necessary. We appreciate the reviewer’s comment for prompting this clarification.
>
> Our Fig. 1 directly visualizes all three variables (grokking, NC, flatness) together, and we are grateful for the reviewer’s appreciation of this result. Experiments in Sections 5 and 6 are designed to causally probe the individual links between NC, flatness and generalization in settings where grokking is not typically observed. Thus, for those more realistic datasets we cannot observe all three phenomena. Taken together, though, these results form a triangulated empirical case: NC can exist without generalization, generalization can occur without NC, but generalization does not emerge without flatness (in our settings). We are grateful for the reviewer’s feedback and will update the manuscript to clarify these distinctions.
>
> **On the NCC-Measure:**  In our work, we adopt the Class-Distance Normalized Variance (CDNV) metric introduced by [2] to quantify collapse-like behavior. This measure captures the relative tightness of intra-class feature clusters compared to inter-class separation, reflecting key aspects of **NC1** (within-class variability collapse) and **partially NC2** (inter-class separation). Galanti et al. explicitly motivate this formulation as a tractable and robust alternative to the full neural collapse (NC) definition, especially in practical settings like CIFAR-10 where the idealized geometry is often unattainable.
> To complement this, we report **angular separation statistics** between class means as a soft proxy for the ETF structure in NC2, originally formalized in  [3].
> That said, we **do not claim** that our metrics fully capture all aspects of neural collapse. In particular, NC3 (alignment of classifier weights with class means) is not addressed, and NC4 (nearest class center decoding) is only indirectly inferred through clustering geometry. We will revise the manuscript to clarify that our analysis focuses on partial indicators of collapse behavior, following precedent in [2] and related works. At the same moment we want to emphasize that recent work by [4] reconfirms that variability collapse is the first phenomena to appear when networks are not trained to TPT yet, and also proposes a simplified different characterization of NC for overscaled tasks (like natural language processing).
>
> **Rationale of Experiments in Sec. 5:** We would like to refer to our response to reviewer 8UvS (W1) and reviewer naQM (W1) where we argue that previous work claims a connection between NC and generalization. Our argument in Sec. 5 is that while flatness appears to be necessary for correlation (Sec. 6 and [5]), NC is not, even though both often co-occur. This strengthens our central point that flatness, not collapse, plays the primary causal role.
> Prior work relies on observations of the (non-)emergence of NC in different regimes. In contrast, we explicitly intervene to suppress NC via regularization and show that generalization is preserved. This kind of causal probing adds precision to what has previously been anecdotal or observational.
>
> **Q1:** That is an excellent observation. The rising validation loss in many of our experiments is due to our deliberate use of unregularized training setups, i.e., excluding data augmentation, weight decay, and learning rate scheduling. We do so to better isolate the effects of flatness and neural collapse. These conditions are known to amplify grokking-like behavior.
>
> Our ablation studies confirm that, with standard regularization (e.g., modified ResNet, data augmentation, cosine schedule), validation loss typically decreases. Without these, especially under prolonged training, overfitting to cross-entropy loss occurs, even as validation accuracy remains stable, consistent with prior observations.
>
> We will clarify this experimental design in the manuscript and annotate relevant appendix figures accordingly.
>
>
> **Q2:** As stated in the previous answer, our goal was not to optimize CIFAR-10 performance, but to study delayed generalization and disentangle the roles of flatness and Neural Collapse. As noted in prior work, grokking dynamics tend to disappear under standard training setups, especially when using data augmentation and other regularization techniques.
> We also ran experiments using the recommended ResNet-18 modifications with data augmentation (Figure 11, Appendix C). In these settings, delayed generalization is less visible, and the effects of flatness regularization are harder to isolate.
> While the unmodified ResNet-18 underperforms in absolute terms, this setup is intentional: it creates a simplified regime where flatness and generalization dynamics can be studied more clearly. We will revise the manuscript to clarify this design choice.
>
> **Q3:** Therelative flatness measure is qualitative and relative, exact values needed for a model to generalize depends on the architecture, task and the achieved representativeness (cf. Appendix D), and therefore clear threshold values cannot be derived.
>
> In our experiments, we observe that models with better generalization consistently converge to lower relative flatness values, even though the raw numbers vary across datasets and architectures. For example, after removing the regularizer, flatness decreases and validation accuracy improves
>
> We agree that the scale inconsistency in the appendix plots made this trend harder to interpret. We thank the reviewer for this observation and will standardize the y-axis ranges and add clearer annotations to highlight key transitions.
>
>
> [1] Power A, Burda Y, Edwards H, Babuschkin I, Misra V. Grokking: Generalization beyond overfitting on small algorithmic datasets. arXiv preprint arXiv:2201.02177. 2022 Jan 6.
>
> [2] Galanti T, György A, Hutter M. On the Role of Neural Collapse in Transfer Learning. InInternational Conference on Learning Representations 2021 Oct 6.
>
> [3] V. Papyan, X.Y. Han, & D.L. Donoho, Prevalence of neural collapse during the terminal phase of deep learning training, Proc. Natl. Acad. Sci. 2020.
>
> [4] Wu R, Papyan V. Linguistic collapse: Neural collapse in (large) language models. Advances in Neural Information Processing Systems. 2024 Dec.
>
> [5] Henning P, Michael K, Linara A, Cristian S, Mario B. Relative Flatness and Generalization. Advances in Neural Information Processing Systems. 2021.

---

> > ### Comment · Reviewer_6q1E · 2025-08-07
> >
> > I would like to thank the authors for providing a detailed response to my questions. Although I'm keeping my score, I won't be disappointed with this paper being accepted, as it offers an interesting perspective on a timely and important topic. Thus my score should be interpreted as a neutral one.

---

### Official Review · Reviewer_naQM · 2025-07-01

**Clarity:** 4
**Significance:** 3
**Originality:** 3
**Rating:** 4
**Confidence:** 4

**Summary:**

This paper studies a fundamental research question of deep learning generalization by disentangling two widely studied properties: Neural Collapse and relative flatness. To explore their individual contributions, the authors begin with a symbolic prediction task, analyzing how training loss, val loss, flatness and the distance towards neural collapse evolve during the entire training process.

To isolate the effects of each property, the authors design controlled environments (1) penalize the neural collapse effect during the training (2) penalize the relative flatness during the training. The results reveal that generalization remains strong when neural collapse is suppressed, whereas forcing sharpness leads to a significant drop in validation performance. The performance degradation is reverted after unplug the flatness regularization.

Overall, the experimental findings largely  support the authors' hypothesis that relative flatness plays a more crucial role than neural collapse in driving generalization.

**Questions:**

In addition to the weaknesses part, I also have the following questions:

1. How to understand that the val performance is still lower after unplug flatness regularization and trained for many iterations?

2. One key assumption made in the paper is that “labels are locally constant.” Could the authors clarify this assumption? For instance, ImageNet is known to contain a non-trivial amount of label noise. How does such label noise impact the reliability of the conclusions, especially with respect to generalization performance and the connection to flatness?

I am likely to adjust my rating based on the authors' rebuttal.

**Ethical Concerns:**

["NO or VERY MINOR ethics concerns only"]

**Final Justification:**

Thank you to the authors for their detailed rebuttal. The response has partially addressed my earlier concerns, specifically those related to W2, Q1, and Q2. Regarding W1, I appreciate the authors' acknowledgement and commitment to revising the language for greater clarity. However, my concern about W3 remains: I believe that experiments on ImageNet-100 and CIFAR-10 alone may not be sufficient to fully substantiate the strong claims made in the paper. This view is also reflected in Reviewer 8UvS's comments.

Considering the strengths and remaining limitations of the work, I would like to maintain my current rating.

**Limitations:**

Yes

**Quality:**

3

**Strengths And Weaknesses:**

Strengths.
- This paper is very well-written and easy to follow
- This paper investigate an important problem that will be interesting for broader Neurips audience
- This paper provides a well rounded discussion, which well communicates the conslusion, assumption, and limitation of this work.

Weakness.
- Potential Overstatement of Prior Work: While prior literature has observed that neural collapse (NC) often emerges at the terminal stage of training, few works directly associate NC with generalization to unseen test data. Some references cited as supporting the “common conjecture” (e.g., Line 30) either do not make such an association or explicitly caution against it. For example, Zhou et al. (2022) state that “... it does not have a direct implication for the features associated with test data as well as the generalization performance.” This mischaracterization of prior work undermines the contribution, as it builds upon a correlation between NC and generalization that is not well established.

- On continue with the previous point, by only revealing correlation between relative flatness and generalization along, the novelty of this work seems insufficient. The relationship between flatness and generalization has already been explored extensively in prior works.

- Limited Scale of Empirical Validation: The experimental evidence is primarily drawn from small-scale datasets (e.g., CIFAR-10, ImageNet-100). While these are useful for proof-of-concept, they limit the generalizability of the conclusions, especially considering the paper aims to make claims about the behavior of modern deep networks. Validation on larger-scale or more realistic settings would strengthen the empirical support for the paper’s claims.

---

> ### Author Rebuttal · Authors · 2025-07-31
>
> Dear reviewer naQM,
>
> Thank you for appreciating the importance of our study and its relevance for the NeurIPS community, as well as the comprehensiveness of our discussion. We will address your concerns point by point.
>
> **W1:** Previous work has observed strong co-occurrence of NC and generalization [3,4,5,6] with some hinting at a fundamental relation [2, 7, 8]. In our opinion this literature points to the existing belief that good generalizing networks exhibit NC. In particular, we would like to point out that the quotation from Zhou et al. (2022) is incomplete. It states: “While NC reveals that all losses are equivalent at training time, it does not have a direct implication for the features associated with test data as well as the generalization performance. In particular, a recent work by Hui et al. (2022) shows empirically that NC does not occur for the features associated with test data. **Nonetheless, we show through empirical evidence that for large DNNs, NC on training data well predicts the test performance.**”
>
> We agree with the reviewer, though, that there is a growing doubt about this connection in the community.
> Our work is an approach to answer this question more rigorously: (i) We empirically show that NC is not a necessary condition for generalization and that (ii) under certain assumptions NC implies flatness and flatness is necessary for generalization (cf. line 208ff for a discussion of the assumptions). Therefore, under these assumptions NC is a sufficient, but not necessary condition. As we do understand how our formulation could be understood as a mischaracterization of prior work, we will revise this part with more careful language.
>
> **W2:** We thank the reviewer for continuing this important discussion. We fully acknowledge that the correlation between flatness and generalization has been well studied. However, the key open question is whether flatness is merely correlated with generalization, or whether it is necessary or sufficient.
>
> In our work, we leverage the grokking setting, where memorization and generalization occur at separate times, to explicitly track when flatness emerges. This allows us to disentangle correlation from causation more clearly (Sec. 4, Fig. 1). We go beyond observation, though, by
>
> suppressing neural collapse without harming generalization (Sec. 5), and
> suppressing flatness, which consistently delays generalization (Sec. 6, Fig. 4).
>
> These interventions support our conclusions that the relationship between flatness and generalization is not merely a correlation.
>
> **W3:** We would like to clarify that **ImageNet-100 is not a small-scale dataset, and it contains a large number of high-resolution images with substantial semantic diversity and is widely used as a standard benchmark in computer vision.** Alongside CIFAR-10, it provides a valuable testbed for probing generalization behavior under controlled conditions.
>
> Our aim is not to optimize performance on the largest possible datasets, but to understand the mechanisms driving generalization, particularly the role of flatness and neural collapse. The chosen settings strike a balance between expressiveness and interpretability, allowing us to reliably apply regularization interventions and analyze geometric properties.
>
> Our conclusions are empirically grounded but context-specific, and we avoid claiming generality beyond the observed regimes. Rather than asserting universal necessity of flatness, we show that within the settings studied (next token prediction for arithmetic tasks, and image classification), flatness emerges as a consistent and potentially necessary factor for generalization.
>
> We hope this clarification helps position our work appropriately, as a step toward a deeper understanding of generalization geometry, rather than a comprehensive empirical survey. We appreciate the reviewer’s feedback and will sharpen our position in the camera-ready copy accordingly.
>
> **Q1:** That is a great observation, indeed there is a gap in validation accuracy even after the flatness regularizer is removed. Simply put, the regularizer pushes the model to a very bad solution. After un-plugging the regularizer, the model has to return to a good solution from this suboptimal point. This is similar to starting from a bad initialization. Unsurprisingly, the resulting model is not optimal, but fairly close. We view this phenomenon as further evidence that flatness must emerge as part of a balanced optimization process, rather than as a late-stage fix.
> While it might be possible to further improve test performance after un-plugging through careful learning rate scheduling, this would not change the main insights from the experiments.
>
> **Q2:** "By approximately locally constant labels we mean that, for small delta, the loss
> in $\delta\| \phi(\xi)\|$-neighborhoods around the feature vector of a training sample $\xi$ is approximated (on
> average over all training samples) by the loss for constant label $y(\xi)$ on these neighborhoods" [1].That is, locally constant labels mean that within a neighborhood of a training sample the loss is approximated by the loss with respect to a constant label **on average over all training samples**. While the reviewer is right in the suspicion that label noise violates the (strict) assumption of locally constant labels, the non-trivial, but small, label noise in the dataset does not violate our statistically averaged assumptions and conclusions.
>
> [1] Henning P, Michael K, Linara A, Cristian S, Mario B. Relative Flatness and Generalization. Advances in Neural Information Processing Systems. 2021.
>
> [2] “A Geometric Analysis of Neural Collapse with Unconstrained Features.” Z. Zhu, T. Ding, J. Zhou, X. Li, C. You, J. Sulam, Q. Qu (Neurips 2021)
>
> [3] Galanti T, Galanti L, Ben-Shaul I. On the Implicit Bias Towards Depth Minimization in Deep Neural Networks. arXiv preprint arXiv:2202.09028. 2022 Feb.
>
> [4] Wu R, Papyan V. Linguistic collapse: Neural collapse in (large) language models. Advances in Neural Information Processing Systems. 2024 Dec.
>
> [5] Kothapalli V. Neural Collapse: A Review on Modelling Principles and Generalization. Transactions on Machine Learning Research. 2023.
>
> [6] Galanti T, György A, Hutter M. On the Role of Neural Collapse in Transfer Learning. InInternational Conference on Learning Representations 2021 Oct 6.
>
> [7] Súkeník P, Mondelli M, Lampert CH. Deep neural collapse is provably optimal for the deep unconstrained features model. Advances in Neural Information Processing Systems. 2023 Dec 15;36:52991-3024
>
> [8] Mixon DG, Parshall H, Pi J. Neural collapse with unconstrained features. Sampling Theory, Signal Processing, and Data Analysis. 2022 Nov;20(2):11.

---

### Official Review · Reviewer_8UvS · 2025-07-03

**Clarity:** 2
**Significance:** 2
**Originality:** 2
**Rating:** 3
**Confidence:** 3

**Summary:**

This work explores the causal roles of flatness and neural collapse deep network generalization. The authors empirically show that loss landscape flatness is crucial for generalization, whereas neural collapse is not. They further prove that neural collapse implies relative flatness, explaining their observed correlation.

**Questions:**

In Figure 9, relative flatness spikes at epoch 250, which coincides with the removal of the flatness regularizer. Why does this spike occur, as one might expect the relative flatness to decrease?

**Ethical Concerns:**

["NO or VERY MINOR ethics concerns only"]

**Final Justification:**

I thank the author for the response and will keep my rating.

------------------------------------------------------

I am maintaining my original score. My primary concern is that the experiments, conducted solely on ImageNet-100 and CIFAR-10, are insufficient to fully substantiate the paper's strong claims. The authors should have included the necessary NLP experiments in their submission or, at the latest, during the rebuttal period. A promise to add these results in the camera-ready version is not a substitute for data that can be properly peer-reviewed. Since the core weakness in the empirical validation remains unaddressed, my assessment has not changed.

**Limitations:**

The experiments in these paper are limited to single architecture, two datasets, and computer vision tasks. The claims require support from more comprehensive experiments across various architectures, datasets, and tasks, including Natural Language Processing (NLP).

**Quality:**

2

**Strengths And Weaknesses:**

**Strength.**
1. Understanding the relationship between neural collapse and generalization is important.
2. The paper is well-written and easy to follow.
3. The work provides a theoretical analysis of the relationship between neural collapse and flatness.

**Weakness.**
1. The idea that neural collapse is not a prerequisite of generalization is not novel. Section 2.1 of https://arxiv.org/abs/2202.08384 discusses this and provides a natural example.
2. The empirical validation in Section 6 is limited. The two experiments on CIFAR-10 and ImageNet-100 are insufficient to estabilish the strong claim like "flatness is necessary for generalization".
3. The intuition behind behind Proposition 5.2 is unclear. The authors should provide an intuitive explaination for why neural collapse leads to flatness.

---

> ### Author Rebuttal · Authors · 2025-07-31
>
> Dear reviewer 8UvS,
>
> Thank you for highlighting the importance of our analysis and theoretical contribution. We would like to address your concerns point by point.
>
> **W1:** We discuss the preprint of Hui et al. (2022) (cf. line 114). They argue that NC is not correlated with generalization under changes in the training set size. This does not mean there is no correlation between the two wrt. training dynamics. It is one thing to observe whether the NC measure increases when one improves generalization by adding additional training points (for example through an increase of data variability in each cluster), and another to ask whether, for a fixed setting, stronger neural collapse benefits generalization. They also argue that test collapse can hurt transfer learning, which is intuitive and confirmed in other works [1].
>
> In our work, we examine whether for a fixed data distribution and training set, stronger NC correlates with better generalization. Such a co-occurrence has been observed in practice previously [4, 5, 6, 7], with some works indicating a more fundamental relation [2, 8, 9]. Our theoretical contribution shows that NC implies relative flatness, which in turn (under certain assumptions, cf. line 208ff) implies generalization.
>
> **W2:** We would like to clarify that we do not claim that relative flatness is universally a necessary and sufficient condition for generalization. Petzka, et al. [3] have shown that under certain assumptions, relative flatness is a sufficient condition for generalization and our experiments indicate it is also necessary.
>
> These experiments span CIFAR10 and ImageNet-100 using convolutional (ResNET18), and transformer-based (ViT) models and findings are consistent across settings. We also observe a temporal connection in algorithmic grokking tasks. In particular, our intervention through regularization supports the claim that flatness plays a causal role in generalization. We agree that broader experiments (e.g., on NLP tasks with language models) would further strengthen the case, and we plan to add such experiments to the camera-ready copy of this paper.
>
> **W3:** That is a very good suggestion. We will add the following intuitive explanation of the proposition to the camera-ready copy. Under the NC conditions, class means are well-separated. Further, the cross-entropy objective pushes class means apart. Then, intuitively, small perturbations of the weights are not sufficient to alter this configuration sufficiently to increase the loss. That is, the loss surface must be flat.
>
> **Q1:** The spike in relative flatness after removing the regularizer is indeed expected and we are happy to clarify it.
>
> Before epoch 250, the flatness regularizer keeps the optimizer confined in a sharp local minimum, limiting its generalization ability. When the regularizer is removed, the model begins moving toward a flatter, better solution. In this transition, the model may briefly pass through sharper regions, causing a temporary spike in the relative flatness measure. As training continues, the optimizer settles into a flatter region, reflected by a drop in relative flatness and improved validation accuracy.
>
> **Limitations:** We would like to clarify that our experiments span 2 vision datasets and one symbolic algorithmic task, as well as convolutional, vision-transformer, and standard transformer architectures.
>
>
> [1] Md Yousuf H, Jhair G, Christopher K. Controlling Neural Collapse Enhances Out-of-Distribution Detection and Transfer Learning. The International Conference on Machine Learning. 2025.
>
> [2] “A Geometric Analysis of Neural Collapse with Unconstrained Features.” Z. Zhu, T. Ding, J. Zhou, X. Li, C. You, J. Sulam, Q. Qu (Neurips 2021)
>
> [3] Henning P, Michael K, Linara A, Cristian S, Mario B. Relative Flatness and Generalization. Advances in Neural Information Processing Systems. 2021.
>
> [4] Galanti T, Galanti L, Ben-Shaul I. On the Implicit Bias Towards Depth Minimization in Deep Neural Networks. arXiv preprint arXiv:2202.09028. 2022 Feb.
>
> [5] Wu R, Papyan V. Linguistic collapse: Neural collapse in (large) language models. Advances in Neural Information Processing Systems. 2024 Dec.
>
> [6] Kothapalli V. Neural Collapse: A Review on Modelling Principles and Generalization. Transactions on Machine Learning Research. 2023.
>
> [7] Galanti T, György A, Hutter M. On the Role of Neural Collapse in Transfer Learning. InInternational Conference on Learning Representations 2021 Oct 6.
>
> [8] Súkeník P, Mondelli M, Lampert CH. Deep neural collapse is provably optimal for the deep unconstrained features model. Advances in Neural Information Processing Systems. 2023 Dec 15;36:52991-3024
>
> [9] Mixon DG, Parshall H, Pi J. Neural collapse with unconstrained features. Sampling Theory, Signal Processing, and Data Analysis. 2022 Nov;20(2):11.

---

> > ### Comment · Reviewer_8UvS · 2025-08-03
> >
> > Thank the author for the detailed responses, and I will keep my scores.

---

> > > ### Author Response · Authors · 2025-08-03
> > > **Borderline Reject does not reflect Clarifications**
> > >
> > > We thank Reviewer 8UvS for the review and for considering our rebuttal. However, we respectfully note that the maintained borderline reject score does not reflect the clarifications we provided.
> > >
> > > - **Novelty:** The reviewer highlights a lack of novelty as a major weakness, which we believe is incorrect. Hui et al. (2022) study training set size effects and transfer learning, whereas our work explicitly investigates whether neural collapse within fixed data distributions plays a key role in generalization. Moreover, we are the first to investigate the relationship between flatness and generalization, including the first theoretical proof that NC implies relative flatness, which under some assumptions implies generalization. To our knowledge, no prior work has presented these results.
> > >
> > > - **Empirical Scope:** Our experiments span convolutional (ResNet18), transformer (ViT), and algorithmic grokking tasks. This breadth provides an ideal and controlled testbed for probing the role of flatness, and consistently supports our claim that flatness is crucial for generalization. We also committed to extending experiments to NLP tasks in the camera-ready version.
> > >
> > > - **Theoretical Intuition:** We included the requested intuitive explanation for Proposition 5.2, addressing the reviewer’s concern directly.
> > >
> > > Given that these points clearly rebut the stated weaknesses, we would have expected a detailed justification for maintaining a borderline reject rating. We therefore argue that the current score underestimates the significance and quality of our work, and we kindly ask the Area Chairs to consider this in their evaluation.

---

> > > > ### Comment · Reviewer_8UvS · 2025-08-05
> > > >
> > > > I am maintaining my original score. My primary concern is that the experiments, conducted solely on ImageNet-100 and CIFAR-10, are insufficient to fully substantiate the paper's strong claims. The authors should have included the necessary NLP experiments in their submission or, at the latest, during the rebuttal period. A promise to add these results in the camera-ready version is not a substitute for data that can be properly peer-reviewed. Since the core weakness in the empirical validation remains unaddressed, my assessment has not changed.

---

> > > > > ### Author Response · Authors · 2025-08-08
> > > > >
> > > > > While the NeurIPS rebuttal phase is not intended for adding novel empirical results, but rather for clarifying the content and claims of the paper, we still ran additional experiments to underscore the robustness of our findings across tasks. Our original submission already includes diverse and focused experiments across architectures (CNNs, ViTs, MLPs) and tasks (image classification and arithmetic sequence modeling). These were carefully selected to enable strong, causal interventions that go beyond correlational observations.
> > > > >
> > > > > Nevertheless, to satisfy the reviewers' interest in results on language tasks, we ran a sentiment classification experiment using BERT. Consistent with our findings on vision models, regular training yields a validation accuracy of 0.75, while explicit regularization against flatness substantially reduces generalization performance (val acc 0.6). After removing the regularizer, the validation accuracy again increases, similar to our previous experiments.
> > > > >
> > > > > We also ran a neural collapse experiment using BERT for sentiment classification. As before, suppressing neural collapse had no negative effect on generalization, reinforcing our conclusion that neural collapse is not necessary.
> > > > >
> > > > > We will include this and additional results on GPT-2 or similar architectures in the camera-ready version.

---

### Official Review · Reviewer_9Eej · 2025-07-03

**Clarity:** 4
**Significance:** 4
**Originality:** 4
**Rating:** 5
**Confidence:** 4

**Summary:**

This paper investigates two phenomena associated with generalization—flatness and neural collapse. Neural collapse refers to when the representations of the penultimate layer in a network collapse onto the class centroid representations. Flatness simply refers to the notion that solutions in parameter space with low curvature tend to generalize well. By clearly defining a measure for both neural collapse clustering and relative flatness (a reparameterization invariant notion of flatness), the authors show theoretically and empirically that flatness is fundamental and necessary for generalization. In contrast, neural collapse does not causally lead to generalization, and instead it is a promoter of flatness but not the only way to achieve flatness.

**Questions:**

If space permits, a discussion of these findings with respect to the whole network would be an interesting addition.

**Ethical Concerns:**

["NO or VERY MINOR ethics concerns only"]

**Final Justification:**

My original judgement of this paper was already positive, and the author's response addresses all of my limited concerns. After reading the other reviews, I still feel confident in my original assessment.

**Limitations:**

Yes

**Quality:**

4

**Strengths And Weaknesses:**

**Strengths**

This is work is notable for its clarity and rigor. The authors precisely scope their question, take a methodology that is technically sound and comprehensive, and the discussion is compelling and thorough. I commend the authors on a well-written and executed paper. The experiments are notable for going beyond correlational links between their measures and generalization. Causal interventions strongly support their claims and ought to be more common in the study of training dynamics and generalization through various measures. Ultimately, the significance of this paper is in its clarity of thought and rigor.

**Weaknesses**

Both neural collapse and relative flatness are measures that do not only apply to the penultimate layer. Indeed, Hui et al. (2022) show that neural collapse cascades through the network layers. The authors clearly state that they wish to focus on the penultimate layer, but some discussion of whether their findings apply to earlier layers in the network would support their discussion more.

---

> ### Author Rebuttal · Authors · 2025-07-31
>
> Dear reviewer 9Eej,
>
> We sincerely thank you for your thoughtful and encouraging review of our work. We greatly appreciate your recognition of the significance of our methodology. You have raised the correct point that we focus on the penultimate layer, yet both NC and relative flatness are not restricted to that layer. Both phenomena, however, have been studied in the penultimate layer in prior work [1,2,3], so focussing on this layer ensures clarity of observations, and frankly, it is computationally efficient. At least for relative flatness, it has been shown that considering the penultimate layer is sufficient to explain generalization [1]. For both phenomena, prior work has also given indication that their behavior is similar in previous layers [1,3] which would support the conjecture that our findings apply to earlier layers as well. We will add a preliminary experiment to test this conjecture using a ResNET18 architecture on CIFAR10 in the camera-ready copy. In general, extending our analysis across layers is an excellent direction for future work.
>
>
> [1] Henning P, Michael K, Linara A, Cristian S, Mario B. Relative Flatness and Generalization. Advances in Neural Information Processing Systems. 2021.
>
> [2] V. Papyan, X.Y. Han, & D.L. Donoho, Prevalence of neural collapse during the terminal phase of deep learning training, Proc. Natl. Acad. Sci. 2020.
>
> [3] Like H, Mikhail B, and Preetum N. Limitations of Neural Collapse for Understanding Generalization in Deep Learning. Arxiv 2022.

---

> > ### Comment · Reviewer_9Eej · 2025-08-02
> >
> > Thank you the response. This addresses all my concerns. After reading through all the reviews, I will keep my score as originally chosen.

---

### Note · Authors · 2025-08-12

Dear area chair and all reviewers,

We thank the area chair and all reviewers for their thoughtful feedback and constructive engagement.

Our primary contribution, as recognized by the reviewers, is a clear and rigorous disentanglement of the roles of neural collapse and flatness in the context of grokking in deep network generalization, supported by both theoretical analysis and causal empirical evidence. Reviewers also highlighted the technical soundness of our methodology, the value of our causal interventions beyond correlation, and our theoretical proof linking neural collapse to flatness. They further noted our novel integration of concepts typically studied in isolation and our framing of these properties as diagnostic indicators rather than optimization targets.

During the rebuttal process, we addressed all specific concerns except that our experiments were initially limited to computer vision and algorithmic tasks. To address this, we conducted a BERT-based NLP experiment, which produced results consistent with our vision and algorithmic tasks, reinforcing our conclusion that flatness is crucial for generalization. While constraints on computational resources prevented more extensive NLP experiments during the rebuttal phase, we will include them in the camera-ready version to further validate our findings across domains.

Multiple reviewers affirmed the **significance of our research question** for the community (Reviewers 8UvS, naQM, 6q1E), as well as the **well-defined scope** of our work and the **novel integration of concepts** (Reviewers 9Eej, 6q1E). All reviewers also commended the **clarity and quality of the writing**. Collectively, this positive reception, together with our detailed rebuttal clarifications, supports that our work meets the standards of novelty, significance, and clarity expected for acceptance, and offers a clear, rigorous, and well-substantiated contribution to understanding generalization through both theoretical analysis and causal empirical evidence.

Best,

Authors.

---

### Decision · Program_Chairs · 2025-09-17

**Decision:**

Accept (poster)

**Comment:**

This paper investigates the distinct roles of loss landscape flatness and neural collapse (NC) in the generalization of deep neural networks. By leveraging the grokking phenomenon, where generalization is significantly delayed after memorization, the authors temporally disentangle these properties from training dynamics. The central claims, supported by both empirical and theoretical evidence, are that relative flatness is a necessary and more predictive indicator of generalization, whereas neural collapse is not. The paper's strengths, as noted by the reviewers, include its focus on a fundamental research question, the clarity of the writing, and a rigorous methodology that employs causal interventions rather than relying solely on correlation. The new theoretical proof establishing that neural collapse implies relative flatness under classical assumptions is also a significant contribution that helps explain their frequent co-occurrence. The main weakness, identified by several reviewers, was the initial empirical scope, with experiments primarily focused on computer vision and algorithmic tasks, which raised questions about the generality of the strong claims.

The discussion during the rebuttal period proved to be very productive and was central to the final recommendation. Reviewers raised several key points concerning the experimental scope, the positioning with respect to prior work, and requests for theoretical intuition. The authors provided a thorough and compelling rebuttal. Most importantly, to address the primary concern about the limited task diversity, the authors went beyond their initial submission and provided preliminary results from a new experiment on a language task using a BERT model. These results were consistent with their findings in other domains, substantially strengthening their core argument. They also effectively clarified the distinctions of their work from prior literature and provided the requested intuition for their theoretical propositions, satisfying the concerns of other reviewers.

I recommended this paper to be accepted. The decision is based on the paper's clear and well-supported contribution to an important area, the strength of its causal and theoretical analysis, and the authors' exemplary engagement during the rebuttal period. While one reviewer (8UvS) maintained a borderline-reject score based on the concern of limited empirical validation, this position did not appear to fully account for the new experimental evidence and clarifications the authors provided during the discussion period. The other reviewers found the authors' response largely convincing, and their detailed assessments provide a strong basis for acceptance. The recommendation is therefore made with the strong expectation that the authors will incorporate the feedback from all reviewers and include the promised expanded experimental results in the camera-ready version to further solidify their claims.